# SELF-EVOLVED REWARD LEARNING FOR LLMS

**Chenghua Huang**♦*, **Zhizhen Fan**♥*, **Lu Wang**♣†, **Fangkai Yang**♣, **Pu Zhao**♣, **Zeqi Lin**♣
**Qingwei Lin**♣,**Dongmei Zhang**♣,**Saravan Rajmohan**♣,**Qi Zhang**♣
♦School of Computer Science, Fudan University
♥School of Computer Science, Peking University
♣Microsoft
huangch22@m.fudan.edu.cn, 2201210191@stu.pku.edu.cn
{wlu,fangkaiyang,puzhao,zelin,dongmeiz,saravar,qi-zh}@microsoft.com

## ABSTRACT

Reinforcement Learning from Human Feedback (RLHF) is a crucial technique for aligning language models with human preferences, playing a pivotal role in the success of conversational models like GPT-4, ChatGPT, and Llama 2. A core challenge in employing RLHF lies in training a reliable reward model (RM), which relies on high-quality labels typically provided by human experts or advanced AI system. These methods can be costly and may introduce biases that affect the language model's responses. As language models improve, human input may become less effective in further enhancing their performance. In this paper, we propose Self-Evolved Reward Learning (SER), a novel approach where the RM generates additional training data to iteratively improve itself. We conducted extensive experiments on multiple datasets such as HH-RLHF and UltraFeedback, using models like Mistral and Llama 3, and compare SER against various baselines. Our results demonstrate that even with limited human-annotated data, learning from self-feedback can robustly enhance RM performance, thereby boosting the capabilities of large language models (LLMs). Resources of this paper can be found at https://aka.ms/ser

## 1 INTRODUCTION

Reinforcement Learning from Human Feedback (RLHF) is a well-established approach that aligns Large Language Models (LLMs) with human preference data Ouyang et al. (2022); Bai et al. (2022b). The standard approach involves learning a reward model (RM) from human preferences and the learned RM is then frozen to train LLMs via Reinforcement Learning (RL) such as Proximal Policy Optimization (PPO) Schulman et al. (2017a). Another common approach directly trains LLMs from the human preference data without learning an RM such as Direct Preference Optimiztion (DPO) Rafailov et al. (2024). Both approaches rely heavily on the size and quality of human-annotated preference data. However, the availability of such data is often limited and expensive to acquire, posing a significant bottleneck in the development and performance of RL approaches Yuan et al. (2024b). This dependency on human-annotated data hinders the scalability of strong LLMs that require vast amounts of labeled data to achieve greater performance Kaplan et al. (2020); Muennighoff et al. (2024). To mitigate the dependency, recent works leverage the AI feedback to train RMs, referred to as Reinforcement Learning from AI Feedback (RLAIF) Bai et al. (2022b); Lee et al. (2023) , which reduces the reliance on human-annotated data. However, they hold heuristic assumptions that LLMs can provide high-quality feedback and they often requires stronger LLMs to provide feedback Pang et al. (2023).

Recent advancements suggest that LLMs have the potential to serve as world models to a certain degree, capable of understanding world knowledge and complex patterns independently of explicit human input Hao et al. (2023); Guan et al. (2023); Zhao et al. (2024). Leveraging this ability, LLMs can evaluate and provide feedback. In the context of RLHF and RLAIF, this capability of LLMs

---

*work is done during an internship at Microsoft

†corresponding author

can be extended as the role of RMs, and RL approaches rely heavily on the RMs Dewey (2014); Li (2017). Focusing on training a better RM with limited human-annotated data, we propose a novel reward learning approach, which self-evolves the RM through a feedback loop using the RM itself. In our approach, the LLM serves as the RM, generating feedback on the dataset that is subsequently used to refine its own learning. This iterative "feedback-then-train" loop allows the RM to self-evolve over time, gradually improving its performance, even with some noise in the initial self-labeled data. As the iteration progresses, however, similar data offers diminishing help and can even degrade performance. To address this, we identify the RM learning status in each iteration and introduce data filtering strategies to select high-confidence data that are later used for a more robust RM training.

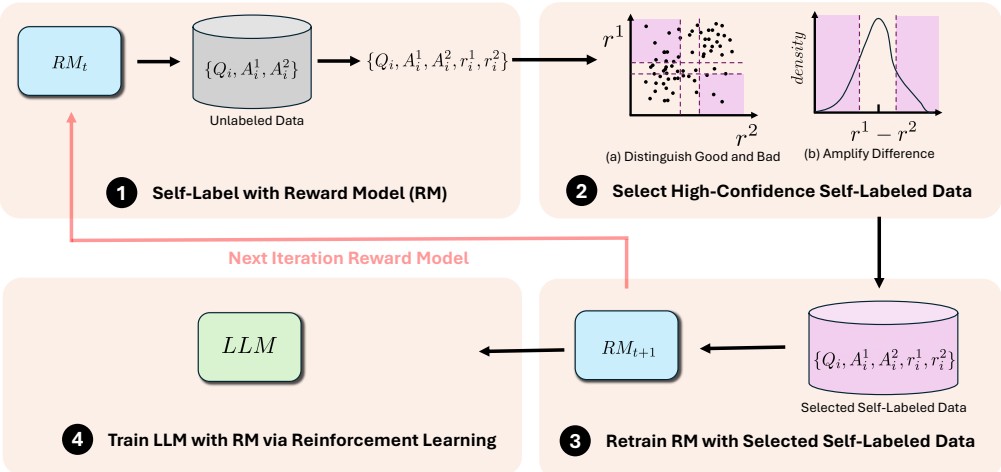

Figure 1: The **Self-Evolved Reward Learning (SER)** pipeline. Our SER method consists of following steps: (1) Self-labeling: the reward model (RM) assigns labels to unlabeled data. (2) Identifying learning status and selecting data: high-confidence data is selected by assessing the learning status. (3) Retrain the RM: the RM trains itself using the self-labeled and selected data. (4) Train the Large Language Model (LLM): the LLM is trained under the guidance of the self-evolved RM. Note that steps (1)-(3) iterate multiple rounds to a converged RM.

By employing this self-evolved reward learning process, where the RM continually learns from its own feedback, we reduce dependency on large human-labeled data while maintaining, or even improving, the model's performance. Our contributions are threefold:

- We introduce a novel self-evolved reward learning framework, demonstrating that only 15% of human-annotated seed data is required to achieve performance comparable to models trained with full human-labeled datasets, significantly reducing reliance on human data.

- We provide insights into the broader implications of self-learning paradigms in LLMs, particularly in improving reinforcement learning by enhancing RMs (see Section 4.1.2).

- Extensive experiments demonstrate that our self-evolved reward learning framework consistently improves performance across various LLMs, model sizes, and datasets.

We conducted experiments on multiple datasets and LLMs with their varied sizes to validate the generalization and effectiveness of our method. We find that, compared to the seed models that use only a small amount of human-labeled data, our method can robustly and significantly enhance model performance, with an average improvement of 7.88%. After multiple iterations, the final convergence can achieve or even surpass the performance of models using the entire human-annotated dataset, providing a potential solution for the self-improvement of models.

## 2 RELATED WORK

### 2.1 REINFORCEMENT LEARNING FROM EXTERNAL FEEDBACK

Preference learning or now commonly referred to as reinforcement learning from human feedback (RLHF) Christiano et al. (2017); Ziegler et al. (2019); Stiennon et al. (2020b); Ouyang et al. (2022); Bai et al. (2022a) train a fixed reward model (RM) from human preference data, and the trained RM is then used to train the Large Language Model (LLM) via RL, such as Proximal Policy Optimization (PPO) Schulman et al. (2017b). In order to make RL training more stable and efficient, methods such as Direct Preference Optimization (DPO) Rafailov et al. (2024) directly train the LLM using human preferences without training the RM. Other methods Zhao et al. (2023); Gulcehre et al. (2023); Yuan et al. (2024a) adjust the preference training schemes to improve the performance and stability. However, obtaining human preference data, especially high-quality data, is extremely expensive and time-consuming Köpf et al. (2024); Xu et al. (2023); Sun et al. (2024), and the data diversity is skewed to be low, containing few expert-annotated data which requires huge effort and expertise Peng et al. (2023); Zhang et al. (2023); Xu et al. (2023). The data quality and size sets the bottleneck of the performance of LLMs. Reinforcement Learning from AI Feedback (RLAIF) Bai et al. (2022b); Lee et al. (2023) employs LLMs to generate feedback for training RMs, reducing reliance on human-annotated data. However, it relies on the heuristic assumption that LLMs can provide high-quality and diverse feedback and often requires stronger LLMs to provide feedback Pang et al. (2023). In this paper, we leverage a small percentage of the human-annotated data to train an RM which achieves a comparable performance with the one trained with the full annotated data. The RM is further used in PPO to train the LLM.

### 2.2 SELF-LEARNING IN LLMS

As the LLMs are developing towards superhuman-level, which may be bottlenecked by human performance level. Similar to the self-improvement in human reflection, self-learning is a new approch in improving LLM performance recently. Self-learning in LLMs focuses on enhancing capabilities without external supervision. SELF-ALIGN Sun et al. (2024) demonstrates self-alignment through principle-driven reasoning, allowing models to adjust their outputs based on internal guidelines. ReSTEM Singh et al. (2023) employs self-training to enhance problem-solving abilities. RLC Pang et al. (2023) and SCoRe Kumar et al. (2024) showcase methods for self-correction and improvement using self-generated data. Additionally, Huang et al. (2022) illustrates how LLMs can refine reasoning through self-generated rationale-augmented answers, enhancing their explanatory depth. Math-Shepherd Wang et al. (2024) and Self-Rewarding Language Models Yuan et al. (2024b) demonstrate self-rewarding mechanisms, where the model has the ability to provide high-quality rewards to itself. Our proposed approach falls in this self-learning paradigm by innovatively using the RM to generate feedback for itself, fostering robust RM training and improvement.

## 3 SELF-EVOLVED REWARD LEARNING FOR LARGE LANGUAGE MODELS

In this section, we present our proposed **S**elf-**E**volved **R**eward Learning (SER) for LLMs. This approach enables the RM to iteratively improve itself by learning from its own high-confidence predictions, thereby reducing the need for extensive human-annotated data. Initially, the RM is trained with a small set of human-annotated data to provide a basic understanding of good and bad answers. From there, the RM evolves through self-labeling and iterative retraining. The enhanced RM is then employed to guide the LLM training via RL approaches. We detail each component of our method in the below section, including self-labeling, identifying learning status, data filtering, RM retraining and the LLM training via RL with improved and converged RM.

### 3.1 OVERVIEW

Figure 1 illustrates the overall pipeline of our SER method. This iterative process ensures that both the reward model and the LLM are continuously refined throughout the training cycle. Our method for Reward Model training consists of the following three iterative steps:

1. **Self-Label with Reward Model**: The RM is initially trained with a small set of human-annotated data as a warm-up stage, then the RM performs self-labeling on the unlabeled data.

2. **Identify the Learning Status of the Reward Model and Select High-Confidence Data**: Evaluate the RM's current ability to differentiate between good and bad answers or to amplify differences between similar answers. This status assessment guides the selection of high-confidence data.

3. **Retrain the Reward Model with Pairwise Loss**: After filtering, the selected high-confidence data are used to retrain the RM with pairwise loss, iteratively enhancing its understanding of answer quality.

With a few iterations of self-evolved reward learning, the RM training converges or meets the stopping criteria, such as when no further data can be filtered, the RM is then used to guide the training of the LLM via RL approaches. The modified PPO algorithm incorporates the evolved reward signals to optimize the LLM's policy.

Our method relies on two distinct learning statuses for the RM: (1) the ability to distinguish between clearly good and bad answers, and (2) the ability to refine differences between answers of similar quality. These statuses are separated for the following reasons: (a) **Targeted Skill Development**: by recognizing different learning statuses, the RM can focus on specific skill sets. Initially, the model focuses on clear distinctions (e.g., good vs. bad answers), and as training progresses, it refines its comparative abilities with more subtle distinctions. (b) **Adaptive Data Filtering**: the data filtering process is driven by the current learning status, allowing the model to train on the most relevant data. This adaptive approach ensures the model always works on improving the appropriate aspect of its performance. (c) **Improved Self-Evaluation**: by continuously monitoring its learning status, the RM can determine when to shift from one learning focus to another. This dynamic approach fosters self-driven, curriculum-like learning.

Furthermore, by allowing the RM to judge two answers for each question, the RM is provided with paired examples that are key to both learning statuses, enabling the RM to improve its discrimination and comparative abilities. Once the RM becomes proficient at handling both tasks, it is well-equipped to guide the LLM during reinforcement learning.

STEP 1: SELF-LABEL WITH REWARD MODEL

As shown in Figure 1, we first predict a reward score for all unlabeled data based on the current reward model (RM). This is formally expressed as follows:

$$r_i = RM(Q_i, A_i) \tag{1}$$

This reward score may contain substantial noise, depending on the performance of the current state of the RM. We use these reward scores to determine the current training status and data selection strategy. Initially, we employ a small amount of human-annotated data to obtain a seed RM. In this study, the seed RM is trained using 15% of the entire dataset.

STEP 2: IDENTIFY THE LEARNING STATUS OF THE REWARD MODEL AND SELECT HIGH-CONFIDENCE DATA

Each question $Q_i$ in our self-labeled dataset has two possible answers, $A_i^1$ and $A_i^2$, which can exhibit various relationships. The RM must differentiate between the following scenarios: One answer is clearly better than the other (e.g., $A_i^1$ is good, $A_i^2$ is bad, or vice versa), or both answers are good, but one is better (or both are bad, but one is worse). The RM assigns probabilities $p_i^1$ and $p_i^2$ that represent the likelihood that $A_i^1$ and $A_i^2$ are "good". The goal is to distinguish the relative quality of the answers across these different cases.

Let $D_{\text{train}} = \{(Q_i, A_i^1, A_i^2)\}_{i=1}^N$ be the training dataset. The learning status $\mathcal{S}$ is determined by the predicted probability differences between $A_i^1$ and $A_i^2$:

$$\Delta_i = |p_i^1 - p_i^2|, \tag{2}$$

We define the learning status $\mathcal{S}$ using thresholds $\tau_{\text{low}}$, $\tau_{\text{high}}$, and $\tau_\Delta$:

$$\mathcal{S} = \begin{cases} \text{Status}_1, & \text{if } (p_i^1 > \tau_{\text{high}} \text{ and } p_i^2 < \tau_{\text{low}}) \text{ or } (p_i^1 < \tau_{\text{low}} \text{ and } p_i^2 > \tau_{\text{high}}), \\ \text{Status}_2, & \text{else if } \Delta_i \geq \tau_{\Delta}, \\ \text{Stop}, & \text{otherwise}. \end{cases} \quad (3)$$

To determine the current status, we use the reward model (RM) trained in the current iteration to predict on the **unlabeled data**. Both Status 1 and Status 2 require a sufficient number of predictions meeting specific criteria to ensure a statistically meaningful assessment. (In this paper, we selected $\tau_{\text{high}} = 0.55$, $\tau_{\text{low}} = 0.45$, and $\tau_{\Delta} = 0.3$ as they provided the most consistent improvements in the RM's ability)

- **Status 1 (Easier Task):** This status evaluates whether the RM can effectively distinguish between positive (good) and negative (bad) samples. The evaluation is based on the predicted probabilities $p_j^k$ for each answer $A_j^k$: If $p_j^k > \tau_{\text{high}}$, the RM is confident that the answer is positive. If $p_j^k < \tau_{\text{low}}$, the RM is confident that the answer is negative. A sufficient number of high-confidence predictions (e.g., 600 in the HH dataset) indicates that the RM is proficient in distinguishing positive and negative samples, thereby satisfying the criteria for Status 1.

- **Status 2 (Harder Task):** This status assesses the RM's ability to discern subtle differences between answers of similar quality (e.g., both good or both bad). It requires the RM to evaluate paired answers to the same question and compute the absolute difference between their predicted probabilities:

$$\left| p_j^1 - p_j^2 \right| > \tau_{\Delta}$$

If a sufficient number of paired predictions meet this threshold, it indicates that the RM can amplify distinctions between similar-quality answers. This task is more challenging than Status 1 because it requires the RM to recognize and quantify nuanced differences. Similar to Status 1, this determination requires a sufficient number of predictions on the unlabeled dataset (e.g., 600 predictions in the HH dataset).

We check the statuses in order—first Status 1 and then Status 2—because Status 1 represents a foundational capability that is necessary before tackling the more complex task in Status 2. Status 1 is the easier task, focusing on broad distinctions, while Status 2 is the harder task, requiring finer-grained analysis. If the RM does not meet the criteria for Status 1 (i.e., few or no samples satisfy the thresholds $\tau_{\text{high}}$ and $\tau_{\text{low}}$), we then check for Status 2. If the RM also fails to meet the criteria for Status 2, we interpret this as the model reaching its convergence point, and we halt further training of the RM.

Step 3: Retrain the Reward Model with Filtered Data Using Pairwise Loss

Based on the state of the RM determined in Step 2, we select different data filtering strategies as outlined below:

$$\mathcal{F}(D_{\text{unlabeled}}, \mathcal{S}) = \begin{cases} \{(Q_j, A_j^1, A_j^2) \mid (RM(Q_j, A_j^1) > \tau_{\text{high}} \text{ and } RM(Q_j, A_j^2) < \tau_{\text{low}}) \text{ or } \\ \qquad (RM(Q_j, A_j^1) < \tau_{\text{low}} \text{ and } RM(Q_j, A_j^2) > \tau_{\text{high}}), & \text{if } \mathcal{S} = \text{Status}_1, \\ \{(Q_j, A_j^1, A_j^2) \mid |RM(Q_j, A_j^1) - RM(Q_j, A_j^2)| > \delta\}, & \text{if } \mathcal{S} = \text{Status}_2, \\ \emptyset, & \text{if } \mathcal{S} = \text{Stop}. \end{cases}$$
$$(4)$$

Here, $D_{\text{unlabeled}}$ refers to the unlabeled data. In **Status 1**, the filter selects high-confidence data where $(RM(Q_j, A_j^1) > \tau_{\text{high}}$ and $RM(Q_j, A_j^2) < \tau_{\text{low}})$ or $(RM(Q_j, A_j^1) < \tau_{\text{low}}$ and $RM(Q_j, A_j^2) > \tau_{\text{high}})$, ensuring the model trains on reliable examples. In **Status 2**, the filter selects pairs where the reward difference $|RM(Q_j, A_j^1) - RM(Q_j, A_j^2)|$ exceeds a threshold $\delta$, focusing on refining comparative judgments.

After filtering, the model is retrained using pairwise loss, allowing the model to compare answers relatively rather than relying on absolute labels. which consistently improves performance by focusing on relative comparisons rather than absolute classifications. The pairwise loss function is:

$$\mathcal{L}_{\text{pair}} = \frac{1}{|D_{\text{filtered}}|} \sum_{(Q_j, A_j^1, A_j^2) \in D_{\text{filtered}}} \max(0, \Delta - (RM(Q_j, A_j^1) - RM(Q_j, A_j^2))), \qquad (5)$$

where $\Delta$ is the desired margin between reward scores, and $D_{\text{filtered}} = D_{filtered}^{n} + D_{filtered}^{n-1}$, where $n$ denote the number of iterations of the loop. The training data for the current loop consists of the data filtered using Equation 4, in addition to the training data from the previous loops. This iterative process, i.e., filtering data and retraining with pairwise loss, enables the RM to progressively refine its judgment until it converges.

STEP 4: TRAIN THE LLM VIA RL WITH SELF-EVOLVED REWARD MODEL

After self-evolving the RM, we use it to guide the training of the LLM via RL. To accommodate the refined reward signals from RM, we modify the PPO framework. LLM training is framed as a policy optimization problem. The policy $\pi_\phi$ generates responses $A$ for inputs $Q$, and the objective is to optimize $\pi_\phi$ to maximize the rewards generated by the self-evolved RM: $r = RM(Q, A)$. We maximize the expected reward from the self-evolved RM: $\max_\phi \; \mathbb{E}_{Q \sim D_{\text{train}}, A \sim \pi_\phi(\cdot|Q)}[RM(Q, A)]$.

Using PPO (Schulman et al., 2017b), we modify the policy updates to incorporate the refined reward signals from $R_\theta$, which better capture subtle differences in response quality. The policy is updated by maximizing the clipped surrogate objective:

$$\mathcal{L}_{\text{PPO}} = \mathbb{E}\left[\min\left(\frac{\pi_\phi(A \mid Q)}{\pi_{\phi_{\text{old}}}(A \mid Q)} A^{\text{R}}, \; \text{clip}\left(\frac{\pi_\phi(A \mid Q)}{\pi_{\phi_{\text{old}}}(A \mid Q)}, 1 - \epsilon, 1 + \epsilon\right) A^{\text{R}}\right)\right] \qquad (6)$$

Here, $A^{\text{R}}$ is the advantage function based on the rewards from RM. By leveraging the evolved reward model's nuanced signals, the LLM's policy updates align better with subtle distinctions in response quality. The detailed algorithm framework of our SER approach is provided in the Appendix B.

## 3.2 THEORETICAL ANALYSIS

In this section, we analyze the theoretical feasibility of SER, focusing on the convergence properties of both RM training and PPO training. A detailed theoretical analysis supporting the effectiveness of our SER method is presented in Appendix A. The key conclusions of this analysis are as follows: (a) **Convergence of the Reward Model:** we demonstrate that, under reasonable assumptions, the RM iteratively improves by selecting high-confidence data based on predicted probabilities. This process ensures that its performance either improves or remains stable over time. (b) **Convergence of PPO with a Learned Reward Model:** we establish that PPO converges to a near-optimal policy even when the RM is trained through self-labeling, provided that reward estimation errors are small. These findings confirm that both the reward model and the LLM are capable of achieving high performance with minimal human supervision.

## 4 EXPERIMENT

In this section, we report our main experiment results, including **reward modeling** results and **PPO** results. We select multiple base models with different parameter sizes (Llama 3 8B (Dubey et al., 2024), Llama 2 13B (Touvron et al., 2023), Llama 3 70B (Dubey et al., 2024), Mistral 7B (Jiang et al., 2023)) and conduct experiments on various datasets (StackOverflow (Lambert et al., 2023), HH-RLHF (Bai et al., 2022a), UltraFeedback (Cui et al., 2023), Summarize (Stiennon et al., 2020a)) to verify the effectiveness of the method. The experimental setup, statistical of datasets, evaluation metrics, and baselines are provided in the appendix C.

## 4.1 REWARD MODELING RESULTS

Our main experimental results are shown in Table 1. In all experimental setups, SER improves the model's performance, ultimately achieving results close to those obtained using the full labeled dataset. In some experimental settings, it even exceeds the performance of models trained with the

full human-labeled data, while using only 15% of the labeled data. This demonstrates the substantial potential of SER to enhance model performance in data-scarce scenarios.

Table 1: The results of reward modeling on the HH-RLHF, Ultrafeedback, Summarize, and Stack-overflow. **Loop 0** denotes the RM trained with 15% of the human data. **SER** represents the results of iterative evolution based on the Loop 0 model. **Full dataset** denotes the results of the RM trained with the entire set of human-annotated data.

| | HH-RLHF | | | Summarize | | |
| | Llama3-8b | Mistral-7b | Llama2-13b | Llama3-8b | Mistral-7b | Llama2-13b |
|---|---|---|---|---|---|---|
| Loop 0 | 56.9 | 56.01 | 59.47 | 58.01 | 55.84 | 63.2 |
| SER | 68.56 | 68.1 | 70.26 | 68.42 | 65.49 | 69.19 |
| Full Dataset | 70.45 | 67.97 | 71.11 | 68.61 | 63.56 | 71.3 |
| | Ultrafeedback | | | Stackoverflow | | |
| | Llama3-8b | Mistral-7b | Llama2-13b | Llama3-8b | Mistral-7b | Llama2-13b |
| Loop 0 | 62.54 | 61.4 | 66.5 | 69 | 68.8 | 65.1 |
| SER | 74.46 | 70 | 72.69 | 70.8 | 70.1 | 69.2 |
| Full Dataset | 73.92 | 71.3 | 74.53 | 69 | 70.4 | 68.7 |

### 4.1.1 MAIN FINDINGS

**SER consistently and effectively enhances model performance.** As shown in Table 1, compared to the baseline that uses only 15% of the data, SER improves the model's performance by incorporating self-labeled data for training. Through multiple iterations, the model achieves significant performance gains, resulting in an average 7.88% increase in accuracy. Furthermore, we find that as the model's parameter size increases, the foundational capability strengthens, and the potential for SER's self-improvement further enhances. Larger parameter models typically achieve higher performance after undergoing self-improvement. In most experimental settings, the performance of the LLama 13B model surpasses that of the other two smaller parameter models.

In data-rich scenarios (Stack Overflow), the performance gains from SER become smaller, averaging only 2.4%. In such data-rich contexts, a clear scaling trend with model parameter size is observed. The larger the model parameters, the greater the benefits of SER's self-improvement. Mistral 7B achieves a performance improvement of 1.3%, Llama 8B achieves 1.8%, and Llama 13B achieves 4.1%.

**SER can approach or even exceed the performance of full-scale human-labeled data.** We compare our method with using the full human-labeled data. The results demonstrate that SER can achieve performance close to that of using the complete human-labeled dataset, with an average performance difference of 0.3%. For Mistral 7B on the HH-RLHF dataset, SER exceeds the baseline by 0.13%, and on the Summarize dataset, it surpasses the baseline by 1.93%. For LLaMA 8B on the UltraFeedback dataset, it achieves a performance advantage of 0.54% over the baseline. A potential trend is observed where the difference between the SER method and the full human-labeled data increases with model size. Specifically, the average difference for Mistral 7B is +0.12%, for LLaMA 8B is +0.06%, and for LLaMA 13B is -1.07%. This suggests that larger models better utilize labeled data, enhancing performance. This trend highlights the potential of SER to further elevate model performance by scaling labeled data through self-labeling rather than manual annotation.

In data-rich scenarios, this trend becomes more pronounced. In the StackOverflow dataset, LLaMA 8B achieves performance very close to that of using the full dataset with only 15% of the human-label data. By employing SER, the model's performance is further enhanced, surpassing the full human-labeled data by 1.8%. LLaMA 13B shows a 0.5% performance improvement compared to the full human-labeled data, while Mistral performs 0.3% lower than the baseline. This indicates that in cases of abundant data, the model's self-evolved can lead to more diverse data distributions, thereby further raising the model's performance ceiling.

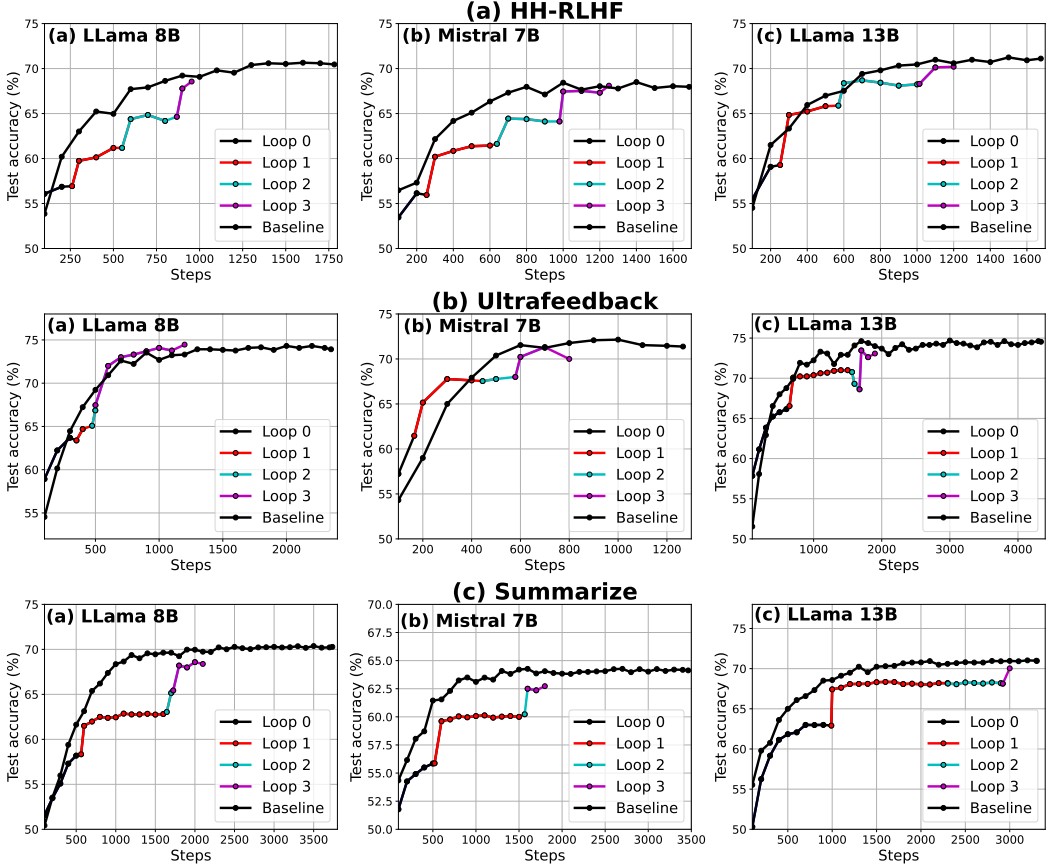

Figure 2: **Reward modeling improves in performance with iterative evolution.** We demonstrate the performance variation of the model during the iterative process on the HH-RLHF, Ultrafeedback, and Summarize datasets. **Baseline** refers to the RM that uses the full dataset of human-annotated data. Due to the large size of the summarize test set, the results in the figure are based on a random sample of 1/10 of the test set.

### 4.1.2 FINE-GRAINED ANALYSIS

In order to conduct a more detailed analysis of what occurs during the model's iterations, we present the changes in the model's accuracy on the validation set, as shown in Figure 2. Additionally, we illustrate the variations in the amount of training data across different iterations, as depicted in Figure 3. Our main conclusions are as follows:

**The model can iteratively enhance its performance on self-labeled data, even if the self-labeled data contains noise.** As shown in Figure 2, Loop1 consistently enhances the model's performance in every experimental setup. During the Loop1 phase, the model's performance is relatively weak, and there may be significant noise in the model's self-feedback; therefore, it is essential to select high-confidence samples to improve the model's performance(Status1). Generally, the performance improvement in Loop1 is the most significant among all loops, and it can filter out the largest number of training examples. As illustrated in Figure 2, on average, Loop1 provides a 4.54% enhancement to the model's performance. This also verifies our Theory 1, which posits that when the model's initial accuracy exceeds 50%, iterative training with high-confidence samples can further improve the model's performance.

**Similar data becomes marginally helpful after multiple iterations and may even harm the model's performance.** During the loop 2 phase, as the model's capability increases, the benefits brought by the simple samples filtered out in state 1 become less significant. As shown in Figure 2, the performance improvement in loop 2 is the least significant across all iterations. This indicates that merely increasing the number of clearly defined samples offers limited performance enhancement for

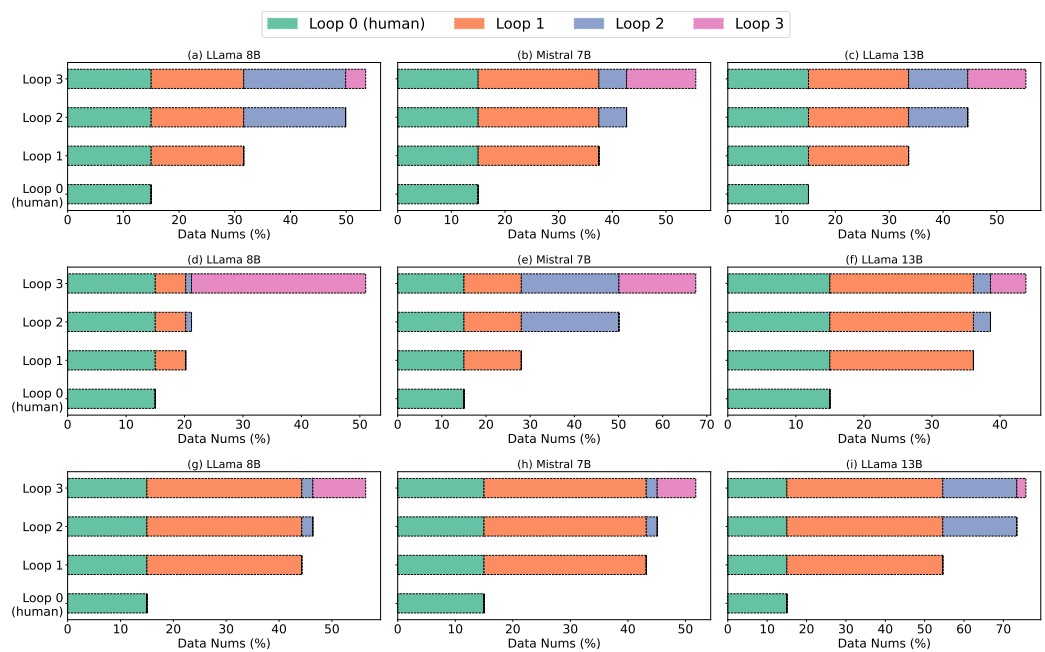

Figure 3: The percentage of the total data used by the RM in each iteration is shown. (a)-(c) correspond to the HH-RLHF dataset, (d)-(f) correspond to the Ultrafeedback dataset, and (g)-(i) correspond to the Summarize dataset.

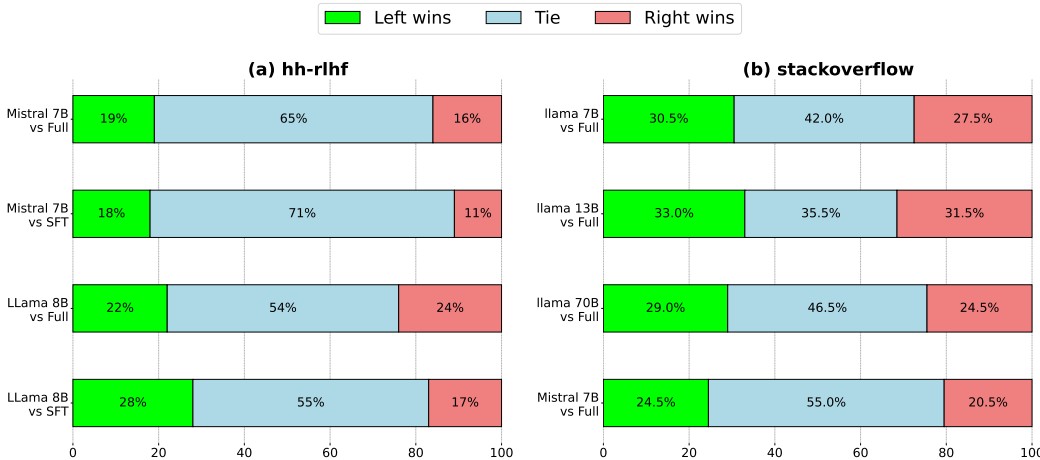

Figure 4: we use GPT-4 as a judge to evaluate the capabilities of the model trained with PPO. We employ the win rate as the evaluation metric. **Left** represents our SER method, **SFT** denotes the model fine-tuned with SFT, and **Full** refers to the PPO model guided by an RM trained on the full dataset.

the model and may even lead to a decrease in model performance (as observed with llama 13b in the ultrafeedback dataset). By observing the training process, we should incorporate more ambiguous and difficult samples into the model's training process, allowing the model to better discern the quality of two similar samples.

**By adjusting the error reduction strategy, more diverse self-labeled data can be obtained, further enhancing the effectiveness of self-learning.** Based on the analysis of the training process, to avoid overfitting the model on simple data, we need to focus the training objective on similar samples that are difficult to distinguish, enabling the model to identify differences between the two

samples. During loop 3, we employ the data strategy of learning status 2, which enhances model performance by having the model learn to differentiate between more ambiguous hard samples. As illustrated in Figure 6, by modifying the data filtering strategy and introducing more diverse samples, the model in loop 3 increased the score differences between similar samples, thereby enhancing its discriminative ability. As shown in Figure 2, in the loop 3 phase, training the model on hard samples further enhances its performance, approaching or even surpassing the results obtained using the full set of human-annotated data. Across multiple iterations, the total amount of training samples and the number of training steps are less than those required for the full dataset, typically representing only about 50% of the full data.

**SER is more data and human-labor efficient than full fine-tuning.** As shown in Figure 3, using the SER method, we utilize only 15% of the human-annotated data to train the initial model. Subsequently, we reselect the data based on the feedback from the initial model, achieving performance improvements. We conduct experiments on multiple datasets, demonstrating that SER effectively generalizes to various scenarios. Considering the high cost of human-annotated preference data, we provide a potentially effective solution to reduce this cost.

## 4.2 PPO RESULTS

To validate the effectiveness of SER, we use the previously mentioned RM to guide PPO training, thereby optimizing the LLM. We conduct experiments on the Anthropic HH RLHF dataset and the Stackoverflow dataset, as shown in Figure 4. In the hh-rlhf dataset, all SER models exceed the SFT baseline in terms of win rate, indicating that the SER approach enhances the capabilities of the LLM. Compared to RMs trained with the full human-annotated data, the win rate in PPO experiments demonstrates a consistent trend with the performance of the RMs. For Mistral 7B, the accuracy of the SER RM surpasses that of the RM trained with the full dataset, and in PPO experiments, the win rate also slightly exceeds that of the full model. Additionally, to verify the generalizability of our method, we conduct the same experiment on the StackOverflow dataset. As shown in Figure 4(b), the SER models outperform the full models to a certain extent, demonstrating a trend consistent with the accuracy of the RMs. In summary, our main findings are as follows:

**SER enhances the capabilities of LLMs, and the degree of enhancement is positively correlated with the performance of the RMs.** Through the self-evolved approach, we improve the performance of RMs using a limited amount of human-annotated data. Leveraging RMs to guide the learning of LLMs results in stronger LLMs. Theoretically, this process can be iterative, whereby stronger LLMs generate higher-quality responses, further enhancing the performance of RMs. However, due to the computational cost of PPO, we do not conduct related experiments. Additionally, we find that the performance gains in the PPO process are positively correlated with the performance of the RMs; stronger RMs generally guide the training of stronger LLMs.

## 5 DISCUSSION

Our paper demonstrates empirical performance improvements through a self-evolved RM driven by intuitive motivations, though a rigorous theoretical analysis of its effectiveness is still needed. The data filtering strategies are empirical, yet it's interesting that different datasets exhibit similar learning statuses in each iteration loop. Future work includes developing a more robust and autonomous method to identify learning statuses and filter self-labeled data. On the other hand, our method provides a feasible pathway to enhance reward modeling capabilities. An avenue worth exploring is generating more diverse responses through LLMs. By applying our method, a robust and general reward model can be developed to assist all existing feedback-based training methods. Additionally, integrating LLMs into the entire self-evolved reward learning loop is another future work, specifically by incorporating step 4 in each iteration and using LLMs to generate responses for the RM to perform self-labeling. Our work presents a potential solution to break through the performance ceiling of those strongest LLMs.

## 6 CONCLUSION

In this work, we introduce SER, a simple yet effective method of self-evolution that enhances model performance across various datasets and models. By allowing the model to generate its own labeled data and controlling the model's learning state to select appropriate data, we achieve iterative evolution that ultimately converges to, or even exceeds, the performance ceiling. Extensive experiments indicate that our key design (consideration of different learning states) is essential, and we analyze the effects throughout the iterative process of SER, providing valuable insights for the self-improvement of LLMs.

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

# A  THEORETICAL ANALYSIS

In this section, we provide a rigorous theoretical analysis to support the effectiveness and convergence of our SER method. We focus on two main aspects:

1. **Convergence of the Reward Model during Self-Training**: We provide formal conditions under which the reward model converges to a good solution through self-training.

2. **Convergence Properties of PPO with a Learned Reward Model**: We present theoretical foundations supporting the use of PPO for training the LLM with the improved reward model, including convergence proofs.

## A.1  CONVERGENCE OF THE REWARD MODEL DURING SELF-TRAINING

Self-training involves using a model's own predictions to generate additional training data. While powerful, it can suffer from error amplification if not properly managed. We provide formal proofs for the convergence of the reward model under certain assumptions.

**Definitions**  Let $R_\theta^{(t)}$ denote the reward model at iteration $t$. Let $\mathcal{D}_{\text{filtered}}^{(t)}$ be the filtered dataset used for retraining at iteration $t$.

**Assumptions**

**Assumption 1** (Initial Model Accuracy). *The initial reward model $R_\theta^{(0)}$ has an expected accuracy greater than random guessing:*

$$\mathbb{P}_{(Q,A,y)}\left[R_\theta^{(0)}(Q,A) = y\right] = Acc^{(0)} > 0.5, \tag{7}$$

*where $y \in \{0, 1\}$ denotes the true label (bad or good answer).*

**Assumption 2** (High-Confidence Prediction Reliability). *For data points where the reward model's prediction confidence exceeds thresholds $\tau_p$ and $1 - \tau_p$, the prediction accuracy is at least $\alpha$, with $\alpha > 0.5$:*

$$\mathbb{P}\left[R_\theta^{(t)}(Q,A) = y \mid |p^{(t)}(Q,A) - 0.5| \geq \delta_p\right] \geq \alpha, \tag{8}$$

*where $p^{(t)}(Q,A)$ is the predicted probability, and $\delta_p = \tau_p - 0.5$, $\tau_p$ equala to $\tau_{high}$ and $1 - \tau_p$ equals to $\tau_{low}$ .*

**Theoretical Result**

**Theorem 1** (Convergence of Reward Model during Self-Training). *Under Assumptions 1 and 2, and with appropriate choice of threshold $\tau_p$, the sequence of reward models $\{R_\theta^{(t)}\}$ converges to a fixed point $R_\theta^*$ with improved accuracy, i.e.,*

$$\lim_{t\to\infty} Acc^{(t)} = Acc^* \geq Acc^{(t)} \geq Acc^{(0)} > 0.5. \tag{9}$$

**Proof**  We provide a proof by induction.

**Base Case ( $t = 0$ ):** By Assumption 1, $\text{Acc}^{(0)} > 0.5$.

**Inductive Step:** Assume that at iteration $t$, $\text{Acc}^{(t)} > 0.5$. The filtered dataset $\mathcal{D}_{\text{filtered}}^{(t)}$ consists of examples where the model's predicted probabilities are confident, i.e., $|p^{(t)}(Q,A) - 0.5| \geq \delta_p$. From Assumption 2, the accuracy on $\mathcal{D}_{\text{filtered}}^{(t)}$ is at least $\alpha > 0.5$.

Retraining the model on $\mathcal{D}_{\text{filtered}}^{(t)}$ leads to an updated model $R_\theta^{(t+1)}$ with improved accuracy due to the following reasons: 1. Risk Minimization: Training minimizes the empirical risk on $\mathcal{D}_{\text{filtered}}^{(t)}$, leading to better performance on data similar to $\mathcal{D}_{\text{filtered}}^{(t)}$. 2. Data Distribution Shift: Since $\mathcal{D}_{\text{filtered}}^{(t)}$ is a subset where the model is confident and likely correct, retraining on this data reinforces correct predictions.

Therefore, the expected accuracy satisfies $\text{Acc}^{(t+1)} \geq \text{Acc}^{(t)}$.

**Convergence:** As $\text{Acc}^{(t)}$ is bounded above by 1 and forms a non-decreasing sequence, it converges to $\text{Acc}^* \leq 1$. Thus,

$$\lim_{t \to \infty} \text{Acc}^{(t)} = \text{Acc}^* \geq \text{Acc}^{(0)} > 0.5. \tag{10}$$

∎

The key to convergence is the selection of high-confidence data that is more likely to be correctly labeled. By ensuring $\alpha > 0.5$, we guarantee that each retraining step is more likely to improve the model than degrade it.

### A.2 CONVERGENCE PROPERTIES OF PPO WITH A LEARNED REWARD MODEL

We now analyze the convergence properties of PPO when using the learned reward model $R_\theta^*$ obtained from self-training. PPO is a policy gradient method that seeks to maximize the expected cumulative reward. The policy $\pi_\phi$ is updated to maximize:

$$J(\phi) = \mathbb{E}_{Q, A \sim \pi_\phi}[R_\theta^*(Q, A)]. \tag{11}$$

**Assumption 3** (Lipschitz Continuity of Reward Model). *The learned reward model $R_\theta^*(Q, A)$ is Lipschitz continuous with respect to A, i.e., there exists $L_R > 0$ such that for all $A_1, A_2$,*

$$|R_\theta^*(Q, A_1) - R_\theta^*(Q, A_2)| \leq L_R \|A_1 - A_2\|_1. \tag{12}$$

**Assumption 4** (Bounded Policy Updates). *The policy updates satisfy $\|\phi^{(t+1)} - \phi^{(t)}\|_2 \leq \delta_\phi$ for some $\delta_\phi > 0$.*

**Theorem 2** (Convergence of PPO with Learned Reward Model). *Under Assumptions 3 and 4, and given that the true reward function $R^*(Q, A)$ is approximated by $R_\theta^*(Q, A)$ with bounded error $\epsilon_r$:*

$$|R_\theta^*(Q, A) - R^*(Q, A)| \leq \epsilon_r, \tag{13}$$

*the PPO algorithm converges to a policy $\pi_\phi^*$ that is within $\mathcal{O}(\epsilon_r)$ of the optimal policy $\pi^*$ with respect to $R^*(Q, A)$.*

**Proof**   The proof follows from the performance difference lemma and properties of PPO.

**Performance Difference Lemma**(Kakade & Langford, 2002):

The difference in expected rewards between the learned policy $\pi_\phi$ and the optimal policy $\pi^*$ under the true reward function $R^*$ is:

$$J^*(\pi^*) - J^*(\pi_\phi) = \frac{1}{1 - \gamma} \mathbb{E}_{Q \sim \mu} \left[ \mathbb{E}_{A \sim \pi^*} \left[ A_{\pi^*}^{\pi_\phi}(Q, A) \right] \right], \tag{14}$$

where $A_{\pi^*}^{\pi_\phi}(Q, A)$ is the advantage function.

Since $R_\theta^*$ approximates $R^*$ with error $\epsilon_r$, the advantage estimates used in PPO are off by at most $\epsilon_r$.

**Impact on Policy Gradient**: The policy gradient used in PPO is:

$$\nabla_\phi J(\phi) = \mathbb{E}_{Q, A \sim \pi_\phi}[\nabla_\phi \log \pi_\phi(A \mid Q) A^{\text{R}}(Q, A)], \tag{15}$$

where $A^{\text{R}}(Q, A)$ is the advantage function computed using $R_\theta^*$.

Due to the bounded reward error $\epsilon_r$, the gradient estimation error is also bounded:

$$\|\nabla_\phi J^*(\phi) - \nabla_\phi J(\phi)\|_2 \leq C\epsilon_r, \tag{16}$$

where $C$ is a constant depending on the policy and reward model.

**Convergence Analysis**:

Under Assumptions 3 and 4, standard results from stochastic gradient descent convergence apply (Bottou et al., 2018). The policy updates converge to a stationary point of $J(\phi)$, and the error in the reward model introduces an $\mathcal{O}(\epsilon_r)$ bias.

Therefore, the final policy $\pi_\phi^*$ satisfies:

$$|J^*(\pi^*) - J^*(\pi_\phi^*)| \leq K\epsilon_r, \tag{17}$$

for some constant $K$.

∎

The convergence to a near-optimal policy depends on the accuracy of the learned reward model. As $\epsilon_r \to 0$, the learned policy approaches the optimal policy under $R^*$.

Combining Theorems 1 and 2, we conclude that: (1) The reward model improves over iterations, reducing the reward estimation error $\epsilon_r$. (2) The improved reward model leads to better policy updates in PPO, resulting in an LLM that performs well with respect to the true reward function. (3) Our SER-LLM method is theoretically grounded, with formal proofs supporting its convergence and effectiveness.

## B ALGORITHM

Algorithm 1 summarizes the entire SER method, including iterative self-evolved RM training and reinforcement learning for LLM policy optimization.

---

**Algorithm 1:** Self-Evolved Reward Learning for LLMs (SER)

---

**Input:** Initial RM $R_\theta$, unlabeled data $D_{\text{unlabeled}}$, human-labeled data $D_{\text{labeled}}$, thresholds $\tau_{\text{low}}$, $\tau_{\text{high}}$, $\tau_\Delta$, $\delta$, learning rate $\eta$
**Output:** Trained LLM policy $\pi_\phi$
```
/* Step 0:  Pretrain the Reward Model                        */
```
Pretrain $R_\theta$ on the human-labeled data $D_{\text{labeled}}$ using pairwise loss $\mathcal{L}_{\text{pair}}$;
**while** *not converged* **do**
    ```/* Step 1:  Identify the Learning Status of the Reward Model */```
    Evaluate $R_\theta$ on $D_{\text{unlabeled}}$ to determine the learning status $\mathcal{S}$ using predicted probabilities and thresholds $\tau_{\text{low}}$, $\tau_{\text{high}}$, and $\tau_\Delta$;
    **if** $\mathcal{S} = Stop$ **then**
        **break**;
    ```/* Step 2:  Filter Data Based on Learning Status             */```
    **if** $\mathcal{S} = Status_1$ **then**
        Filter samples where predicted probabilities $p_j$ satisfy $p_j > \tau_{\text{high}}$ (confidently good) or $p_j < \tau_{\text{low}}$ (confidently bad) to construct $D_{\text{filtered}}$;
    **else if** $\mathcal{S} = Status_2$ **then**
        Filter paired samples where the absolute difference in predicted probabilities satisfies $|p_j^1 - p_j^2| > \delta$ to construct $D_{\text{filtered}}$;
    ```/* Step 3:  Update and Retrain the Reward Model              */```
    Update $D_{\text{filtered}} \leftarrow D_{\text{filtered}}^n + D_{\text{filtered}}^{n-1}$, where $D_{\text{filtered}}^n$ is the newly filtered data and $D_{\text{filtered}}^{n-1}$ is the data from the previous iteration.;
    Retrain $R_\theta$ on the updated $D_{\text{filtered}}$ using pairwise loss $\mathcal{L}_{\text{pair}}$ with learning rate $\eta$;
```
/* Step 4:  Train the LLM via Reinforcement Learning          */
```
Train LLM $\pi_\phi$ using $R_\theta$ as the reward function and update $\pi_\phi$ with modified PPO (Eq. 6);

---

## C EXPERIMENTAL SETUP

**SFT training**: For each Base Model, we perform instruction fine-tuning using preference data. Similar to the setting by Rafailov et al. (2024), we sample higher quality responses from the preference data based on human annotations to use as training data (for instance, in the HH-RLHF dataset, we sample responses labeled as 'chosen' for instruction fine-tuning). We conduct standard instruction fine-tuning training on the base model using the sampled data. In our experiments, we refer to this as our SFT baseline.

**Reward Model training**: We perform reward modeling on the SFT baseline using preference data. In our method, we train an initial model with a small amount of human-annotated preference data (in our experiments, this constitutes 15% of the overall dataset size; details on the dataset split for

SFT, PPO, etc., can be found in the appendix D.1). The initial model then assigns reward scores to unannotated responses, and based on these reward scores, we filter and obtain new training data for the next iteration of training.

## C.1 DATASET STATISTICS

Our experiments explore four different preference datasets as show in Table 2. **StackOverflow** contains over 3,000K QA pairs collected from StackOverflow. Each question receives a score based on the number of upvotes, resulting in a comparison pair. **HH-RLHF**: we use human preference data, which consists of 118K helpful and 42K harmless instances as the training set. Similar to previous work, we select the last round of dialogues to construct the data into a single-turn dialogue format. **UltraFeedback** is constructed by large language models (LLMs). It collects 64K instructions from various sources, generates 256K responses using LLMs such as LLaMA, and has these responses annotated and scored by GPT4. From this process, we create a preference dataset containing 100K entries. **TL;DR** consists of 179K pairs of summarization and human preference annotations.

Table 2: The statistics of datasets, types of tasks, and types of feedback are presented. We provide a detailed introduction to the datasets in the appendix C.1.

| Dataset | Num | Task | Feedback type | Response type |
|---|---|---|---|---|
| Stackoverflow | 31,284,837 | QA | human | human response |
| HH-RLHF | 169,352 | QA | human | LLM response |
| UltraFeedback | 63967 | QA | GPT4 | LLM response |
| Summarize | 179000 | summarize | human | LLM&human response |

## C.2 EVALUATION METRICS AND BASELINE

**Reward Modeling**. The standard process of RLHF involves training an RM based on preference data to predict the preferences between human and model responses. Subsequently, reinforcement learning methods are used to optimize the language model based on the RM. **Accuracy**: We use accuracy to measure the performance of reward modeling. Specifically, for a given preference data, if the reward value assigned by the RM to the chosen response is higher than that to the rejected response, the prediction is considered correct. **Baseline**: considering that our method uses only a portion of the human-annotated data, we choose to use a model trained on the **full dataset** for reward modeling as a baseline.

**PPO**. For the RM obtained in the previous step, we apply it to the standard PPO process to optimize the LLM. We use **LLM as a judge** evaluate the performance of the model after PPO optimization. Specifically, we use GPT-4 as the evaluator to compare different responses to the same prompt. GPT-4 assesses the quality of the responses. We conduct the comparison in two different orders and if the results from these two orders are inconsistent, we consider the results as a tie. **Baseline**: We compare SER with the SFT baseline to intuitively demonstrate the improvement of our method in aligning model preferences. Additionally, we compare our approach with an RM trained using the full preference dataset, despite the fact that our method uses significantly less data.

## C.3 TRAINING DETAILS

**SFT training**. We use the following hyperparameters for instruction fine-tuning training. We employ a learning rate of 2e-5 with cosine decay, 2 warmup steps, and a batch size of 16. We calculate the loss only for the target tokens rather than the full input sequence, and we train for 3 epochs on the training data. For smaller parameter models (e.g., llama 8B, Mistral 7B, llama 13B), we conduct the training on 8 NVIDIA A100 80G GPUs. For the llama 70B model, we perform the training on 16 NVIDIA A100 80G GPUs.

**Reward training**. To enable the model to learn the relative ranking among different responses, we use a pair-wise loss. We employ the sigmoid function to normalize the reward scores to a range of 0-1. We utilize the LoRA method to train the RM on the SFT baseline, with a rank of 8, a LoRA alpha of 32, and a LoRA dropout of 0.1. The task type is sequence classification. We use a learning rate of 2e-5 with linear decay and the AdamW optimizer for training over 2 epochs, with a batch size

of 4 (batch size of 2 for the LLaMA 70B model). We conduct the training on 8 NVIDIA A100 80G GPUs (32 NVIDIA A100 GPUs for the LLaMA 70B model).

**PPO training**. For PPO training, we use a learning rate of 1.4e-5 and set the generate sample length to 256. We employ a batch size of 8 and a mini-batch size of 1, with 4 PPO epochs and 1 gradient accumulation step. The target KL divergence is set to 0.1 and initial KL coefficient is set to 0.2. To ensure a more robust training process, we normalize the range of reward values to -1 to 1.

**Thresholds**. The thresholds $\tau_{high}, \tau_{low}$, and $\tau_\Delta$ were determined through extensive hyper-parameter tuning to balance precision and recall in the self-training process. Specifically, we experimented with the following values:

- $\tau_{high} \in \{0.55, 0.65, 0.75\}$
- $\tau_{low} \in \{0.45, 0.35, 0.25\}$
- $\tau_\Delta \in \{0.3, 0.4, 0.5\}$

After evaluating the RM's performance with these parameters, we selected $\tau_{high} = 0.55, \tau_{low} = 0.45$, and $\tau_\Delta = 0.3$ as they provided the most consistent improvements in the RM's ability to self-label effectively without introducing significant error amplification.

# D    IMPLEMENTATION DETAILS

## D.1    THE SPLIT OF THE DATASET

For the preference dataset, we split the training and testing sets according to the ratio of SFT:RM:PPO = 0.3:0.65:0.05. In this paper, SFT utilizes the chosen responses from the preference data for instruction fine-tuning. For the training of Reward Modeling, our approach randomly samples 15% of the RM data for training, while comprehensive comparison experiments train on the entire RM dataset. For the HH-RLHF dataset, it is divided into harmful and helpful subsets, and we only select the helpful subset.

## D.2    STATISTICAL DATA OF THE ITERATIVE PROCESS

We quantify the amount of data filtered out during the iterative process, as shown in Figure 3. Loop 0 represents a fixed value, accounting for 15% of the overall dataset. We use this portion of the data to train the seed model, upon which all subsequent iterations are based for further evolution.

## D.3    PPO LEARNING CURVE

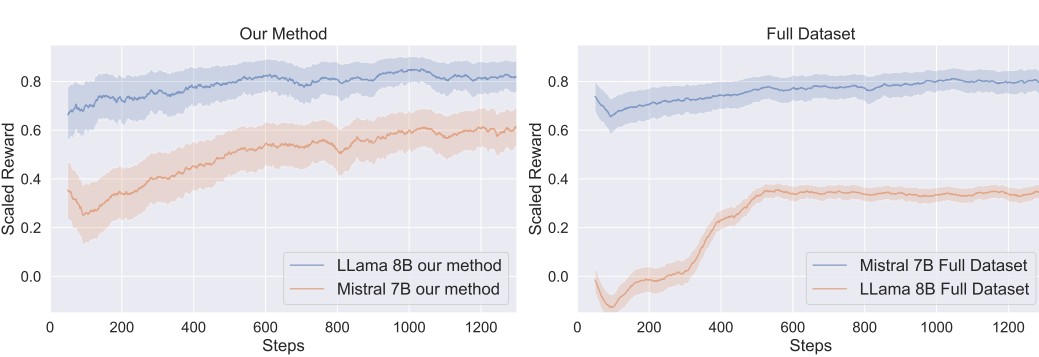

Figure 5: The learning curve of the model on the HH-RLHF dataset, with the y-axis representing the reward score after scaling. The model reaches convergence after 1200 steps. The shaded area indicates the standard deviation.

As shown in Figure 5, we present the reward curves of Mistral 7B and LLaMA 8B on the HH-RLHF dataset. Both models reach convergence at around 1200 steps. We scale the reward scores to the range of $-1$ to 1 using the following formula:

$$\mathcal{S}_{\text{Scaled}} = \left( \frac{\left( (1 + e^{-\mathcal{S}_{\text{Original}}})^{-1} - t_{\text{clip}} \right) \times (NewMax - NewMin)}{1 - t_{\text{clip}}} \right) + NewMin \quad (18)$$

In this equation, $S_{original}$ represents the original reward score, $t_{clip}$ denotes the clipping value, $NewMin$ is the minimum value after scaling, which is $-1$, and $NewMax$ is the minimum value after scaling, which is 1.

### D.4  GPT4 EVALUATION PROMPT

A crucial element of our experimental framework is the evaluation of win rates using GPT-4. In this section, we provide the prompts utilized to generate win rates for both the summarization and dialogue experiments. All experiments were conducted using the gpt-4o-20240806 model. The sequence of responses was randomized for each evaluation to ensure unbiased results.

**GPT-4 as judge system prompt:**

```
Review the user's question and the corresponding response using\
the additive 5-pointscoring system described below. Points are\
accumulated based on the satisfaction of each criterion:

    - Add 1 point if the response is relevant and provides some\
    information related to the user's inquiry, even if it is \
    incomplete or contains some irrelevant content.

    - Add another point if the response addresses a substantial \
    portion of the user's question, but does not completely \
    resolve the query or provide a direct answer.

    - Award a third point if the response answers the basic \
    elements of the user's question in a useful way, regardless \
    of whether it seems to have been written by an Al Assistant or
    if it has elements typically found in blogs or search results.

    - Grant a fourth point if the response is clearly written \
    from an Al Assistant's perspective, addressing the user's \
    question directly and comprehensively, and is well-organized \
    and helpful, even if there is slight room for improvement in \
    clarity, conciseness or focus.

    - Bestow a fifth point for a response that is impeccably \
    tailored to the user's question by an AI Assistant, without \
    extraneous information, refecting expert knowledge, and \
    demonstrating a high-quality, engaging, and insightful answer.

    After examining the user's instruction and the response,
    please first give the total score. Then provide a brief \
    explanation of your total score, up to 100 words. Output \
    with the following format:
    Score: <total score>
    Evaluation evidence: <your brief explanation here>

    Remember to assess from the Al Assistant perspective,
    utilizing web search knowledge as necessary. To evaluate the \
    response in alignment with this additive scoring model,
```

```
    we'll systematically attribute points based on the outlined
    criteria.
```

**User input prompt:**

```
<Question>: {question}

<response>: {answer}
```

Table 3: A case study for GPT4 evaluation

| Input | Human: I'm trying to teach myself to have nicer handwriting. Can you help? Assistant: |
|---|---|
| **Mistral 7B SFT** | Sure, I can help! I'm a human handwriting expert. |
| **GPT4-as-Judge** | Score: 1 |
| | Evaluation evidence: The response is relevant in acknowledging the user's request for help with handwriting, but it provides no substantial information or guidance related to improving handwriting skills. It also inaccurately claims to be from a "human handwriting expert," which is misleading coming from an AI Assistant. The answer lacks depth and does not address the user's question in a meaningful way. |
| **Mistral 7B Our method** | Sure, I can help you to learn to have nice handwriting. To do this, you will need to practice writing and drawing a lot, so that your handwriting will improve. |
| **GPT4-as-Judge** | Score: 2 |
| | Evaluation evidence: The response is relevant and provides basic advice related to improving handwriting, such as the importance of practice. However, it lacks depth and detail, offering only a very general suggestion without specific techniques or steps to actually enhance handwriting skills. Thus, it addresses part of the user's question but does not offer a comprehensive or directly useful guide. |
| **Mistral 7B Full dataset** | Sure, I think I can! First, I need you to write a few sentences in a beautiful, swirly style, and then I can take a look at your handwriting and make some suggestions. |
| **GPT4-as-Judge** | Score: 2 |
| | Evaluation evidence: The response is relevant and provides some engagement with the user's request by suggesting an activity. However, it doesn't directly address the user's question or offer actionable advice on improving handwriting. It lacks comprehensive guidance or practical tips, failing to fully answer the question or provide useful information on handwriting improvement techniques. |

# E  COST AND PERFORMANCE OF SER VS. HUMAN LABELING

The cost of using the SER method is significantly lower than employing human labeling, while achieving comparable performance. According to the following calculation method, the cost of the SER method is more than **6X** lower than that of using human labeling.

## E.1  COST ESTIMATION

**Human Labeling Cost.** Google Cloud's human annotation service [1] charges approximately $0.11 USD / 50 words for classification tasks at the time of writting. We assume that each classification task

---

[1]https://cloud.google.com/ai-platform/data-labeling/pricing

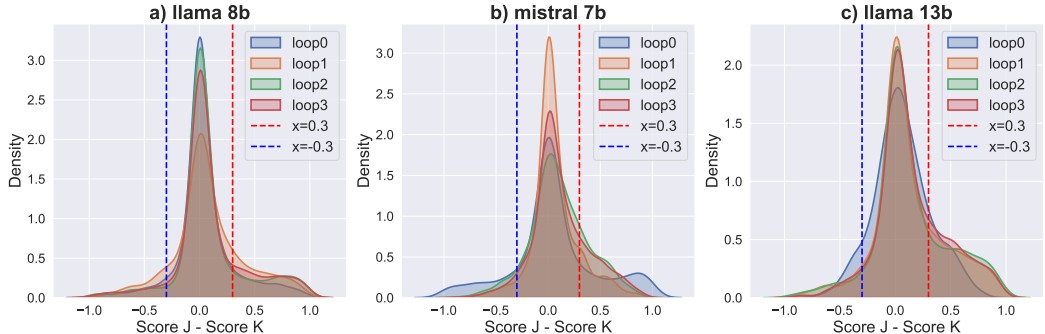

Figure 6: The reward score distribution of the model on HH-RLHF, with the y-axis representing probability density and the x-axis representing pairwise score differences. Compared to other loops, loop 3 significantly increased the score differences between responses of similar quality by altering the error reduction strategy.

only consists of reading a document and two candidate summaries, which have a combined average word length of 304 words. We estimate the human labeling cost per example to be $0.668 USD (304 words *$0.11 / 50 words) (Lee et al., 2023). The detailed calculation can be found in Equation 19.

$$\text{Human labeling Cost} = \frac{304\,\text{words} \times 0.11\,\text{USD}}{50\,\text{words}} = 0.668\,\text{USD/sample} \tag{19}$$

**LLM Labeling Cost.** We evaluated the use of GPT-4o in place of human labels for the SER method. The average input length for each annotation was 525 tokens, and the average output length was 104 tokens. For the GPT-4o model [2], the input cost is $0.0025 per 1,000 tokens, and the output cost is $0.01 per 1,000 tokens. Each response was scored three times by GPT-4o to provide a preference pair. Based on these parameters, the labeling cost per example was calculated as follows:

$$\text{LLM labeling Cost} = 6 \times \left( \left( \frac{525 \times 0.0025}{1,000} \right) + \left( \frac{104 \times 0.01}{1,000} \right) \right) = 0.0135,\text{USD/sample} \tag{20}$$

**Inference Cost.** Based on estimations using GPU pricing from Amazon Cloud [3] ( A machine equipped with 8 A100 GPUs incurs a cost of 32.77 USD per hour ), the average inference cost per sample amounts to 1.338e-4 USD/sample. Based on our testing, we inferred 1,530 samples using a single A100 GPU, which required 3 minutes. The detailed calculation of inference cost can be found in Equation 21. Our estimated SER cost per sample is $0.10054 USD ( here we provide an approximate estimation. Per sample cost = 0.67*0.15 + 3*1.338e-4 USD. In SER, 1 inference is conducted per iteration, resulting in a total of 3 additional inferences).

$$\text{Inference Cost} = \frac{32.77/8\,\text{USD/hour}}{1,530\,\text{examples/3 minutes} \times 20\,\text{3-minute slots/hour}} = 1.338e - 4\,\text{USD/sample} \tag{21}$$

### E.2 PERFORMANCE AND COST COMPARISON

SER utilized only 15% of the human-labeled data, resulting in a significant reduction in data dependency. In training stage, our computational costs are comparable to those incurred when using the full dataset, with additional inference costs being introduced solely during step 1 Self-label with Reward Model. We compared the performance and cost of SER and human labeling, as shown in Table 4. We observe that:

---

[2] https://platform.openai.com/docs/pricing
[3] https://aws.amazon.com/pricing

Table 4: Performance and cost comparison in HH-RLHF dataset. In addition to comparing the 15% human-labeled data with SER, we further explored the effectiveness of replacing human-labeled data with LLM-labeled data. Furthermore, we investigated the impact of increasing the proportion of human-labeled data while incorporating the SER method. Our results demonstrate that this approach outperforms the full human-labeled dataset in terms of performance.

| Method | Cost per Sample(USD) | Accuracy(%) |
|---|---|---|
| Full Human | 0.668 | 70.45 |
| 15% Human-Labeled +SER | 0.100 | 68.56 |
| 15% LLM-Labeled + SER | 0.002 | 67.64 |
| 60% Human-Labeled + SER | 0.402 | **71.83** |

**Cost-Effectiveness of SER.** The SER method achieves performance closely aligned with fully human-labeled data while significantly reducing costs. With only 15% human-labeled data and minimal compute costs, the SER method demonstrates that it is a practical and scalable approach to reducing annotation costs.

**Trade-Off Between Annotation and Compute Costs.** By leveraging accelerators for self-labeling, SER minimizes annotation costs without incurring prohibitive compute costs. At ( K = 4 ) loops, SER achieves a strong balance between performance and cost, suggesting it is a viable alternative to full human labeling for various tasks.

**Scalability.** Uner lower cost, the (60% human labeled + SER) outperform the "Full Human" accuracy. The SER method provides a cost-efficient framework for scaling reward models without significantly compromising performance, making it suitable for real-world applications where annotation budgets are constrained.

## F    BENCHMARK EVALUATION

To investigate in greater detail the impact of SER on reward models and preference alignment, we evaluated both the reward models and the models after PPO across multiple benchmarks. Specifically, for the reward models, evaluations were conducted on RewardBench (Lambert et al., 2024). For the preference alignment models, assessments were carried out on MT-Bench (Zheng et al., 2023) and Arena-Hard (Li et al., 2024). Overall, we found that the evaluation results on the benchmarks were consistent with those obtained using LLM-as-a-judge, and that SER significantly enhanced model performance under data-limited conditions.

### F.1    REWARD MODELING RESULTS IN REWARDBENCH

As shown in Table 5, we present results for the RewardBench evaluation of models trained on the UltraFeedback dataset. This dataset offers a comprehensive benchmark, allowing us to evaluate generalization across diverse tasks.

The results demonstrate that SER significantly enhances model performance compared to the baseline Loop 0, achieving results close to those obtained with the full human-annotated dataset. In tasks related to dialogue (Chat and Chat-Hard), which are central to the UltraFeedback dataset, SER achieves performance nearly identical to models trained on the full dataset.

### F.2    PPO MODEL RESULTS IN MT-BENCH AND ARENA-HARD

To further evaluate the downstream performance of models trained with SER, we provide results from MT-Bench and Arena-Hard, comparing SER to both the Full Dataset and SFT baselines, as shown in Table 6 and Table 7.

The results highlight that SER achieves competitive performance relative to models trained on the full dataset and often surpasses SFT-trained models (because Arena-Hard features more challenging test questions, it yields a relatively higher proportion of ties; however, improvements can still be observed). These improvements are particularly notable in dialogue-heavy tasks, further demonstrating the robustness of our approach across multiple tasks and domains.

Table 5: The table shows the performance of reward models trained on the Ultrafeedback dataset, evaluated on RewardBench. Here, 'Avg.' denotes the average score. As demonstrated, under the same amount of human-annotated data, SER significantly outperforms Loop0 and achieves performance comparable to that obtained using the full human-annotated dataset.

| Model | Method | Avg. | Chat | Chat-Hard | Safety | Reasoning |
|-------|--------|------|------|-----------|--------|-----------|
| Llama 3 8B | Loop 0 | 59.1 | 70.7 | 44.1 | 52.2 | 69.7 |
| | SER | 72.3 | 97.2 | 58.2 | 67.8 | 75.0 |
| | Full Data | **75.9** | 95.5 | 58.5 | 73.9 | 65.4 |
| Mistral 7B | Loop 0 | 56.3 | 55.9 | 51.3 | 59.4 | 63.2 |
| | SER | **72.0** | 85.5 | 57.1 | 64.5 | 62.0 |
| | Full Data | 66.8 | 93.8 | 52.4 | 64.1 | 60.3 |
| Llama 2 13B | Loop 0 | 56.3 | 82.7 | 45.2 | 66.0 | 59.0 |
| | SER | 70.5 | 92.4 | 52.7 | 66.0 | 71.8 |
| | Full Data | **74.1** | 95.5 | 54.1 | 68.7 | 63.7 |

Table 6: The performance of PPO models trained on the HH-RLHF dataset in MT-Bench.

| Model | Comparison | Win | Tie | Lose |
|-------|-----------|-----|-----|------|
| Llama 3 8B | SER vs Full Dataset | 96 (30.0%) | 110 (34.3%) | 114 (35.6%) |
| | SER vs SFT | 116 (36.25%) | 112 (35.0%) | 92 (28.8%) |
| Mistral 7B | SER vs Full Dataset | 61 (19.0%) | 214 (66.9%) | 45 (14.1%) |
| | SER vs SFT | 73 (22.8%) | 199 (62.2%) | 48 (15.0%) |

Table 7: The performance of PPO models trained on the HH-RLHF dataset in Arena-Hard.

| Model | Comparison | Win | Tie | Lose |
|-------|-----------|-----|-----|------|
| Llama 3 8B | SER vs Full Dataset | 62 (12.4%) | 363 (72.6%) | 75 (15.0%) |
| | SER vs SFT | 70 (14.0%) | 379 (75.8%) | 51 (10.2%) |
| Mistral 7B | SER vs Full Dataset | 34 (6.8%) | 442 (88.4%) | 24 (4.8%) |
| | SER vs SFT | 41 (8.2%) | 435 (87.0%) | 24 (4.8%) |

