# OpenReview forum: "SELF-EVOLVED REWARD LEARNING FOR LLMS"
_ICLR.cc/2025/Conference — ICLR 2025 Poster_

### Official Review · Reviewer_eZmA · 2024-10-23

**Soundness:** 1
**Presentation:** 2
**Contribution:** 2
**Rating:** 5
**Confidence:** 4

**Summary:**

This paper proposes self-evolved reward learning where the reward model is used to obtain more training data for iterative improvement. The algorithm is straightforward: (1) generate two responses per prompt and label using RM, (2) only keep response pairs where reward difference is large -- there are two filtering strategies (Status 1 and Status 2) and the authors alternate between them, (3) train RM with the filtered data; then, iterate the above. Finally, use RM for LM training. Findings: Only 15% human-annotated data is needed so that the new model archives performance similar to training on the entire human-annotation dataset; experiments are done on a few models (7B to 13B).

**Strengths:**

The self-improvement paradigm could save lots of resources.

I appreciate the fact that the authors not only tested on reward modeling benchmarks but also LMs (by doing PPO).

Experiments are done on multiple models and multiple model sizes.

**Weaknesses:**

On baselines:
- It’s unclear how specifically the authors trained “Loop 0” and “Full dataset” baseline models. What is the loss function? How are hyperparameters tuned?
- Have the authors tried regular scalar-valued reward functions as baselines?

On eval: Have the authors tested on common benchmarks like RewardBench (so we can easily compare to other scalar and generative RMs)?

In real LLM training, the same dataset would always contain examples of multiple domains. In the same dataset, lots of examples could be on simple helpfulness prompts, and lots of examples are on coding/reasoning prompts. In this case, how would thresholding work? Would the model first learn simple prompts and then coding/reasoning/complicated prompts? Or would the model give up on coding/reasoning/complicated prompts?

Eq. (3) can be clearer:
- It looks like Status 1 and Status 2 can be achieved at the same time? In that case, does S=Status1 or Status2?
- Do we have a different S at every i?
- Do we need to recompute Delta_p (avg through every i) after each gradient step?
- When would the model hit “Stop”? What if the “otherwise” case never happens? An explanation in natural language would be great.

Details are lacking on data generation/filtering
- How specifically are responses generated?
- What percentage of data are filtered at different point in training?
- Are there more conclusive evidence that alternating between status are beneficial? How are hyperparameters tuned -- based on what metrics?


Other misc issues:
- Eq. (4): missing right bracket
- Table 1 caption mentions “Our method” but I don’t see “Our method” in the table. Do you mean SER?
- Serious misuse of \citet and \citep. Most of the citations on the first page and the second page, for example, should be in \citep.

———

UPDATE: increased score to 5 but still concerned about paragraph 3 (and hopefully more detailed result on paragraph 2) above.

**Questions:**

Are Llama models base models or instruction-tuned models?

---

> ### Author Response · Authors · 2024-11-23
> **Response to Comment1 and Comment2**
>
> **Comment 1:** On baselines
>
> **Response to Comment 1:**
>
> **It’s unclear how specifically the authors trained “Loop 0” and “Full dataset” baseline models. What is the loss function? How are hyperparameters tuned?**
>
> Thank you for your comment. To clarify, our experimental setup, including details on training and hyperparameter tuning, is described in **Appendix C (lines 818–824)** . Below, we summarize the key points addressing your concerns.
>
> **1.** **Loss Function**
>
> We use the same pairwise loss function across all models, as defined in Equation 5:
>
> $L_{\text{pair}} = \frac{1}{|D_{\text{filtered}}|} \sum_{(Q_j, A_j^1, A_j^2) \in D_{\text{filtered}}} \max(0, \Delta - (RM(Q_j, A_j^1) - RM(Q_j, A_j^2)))$
>
> where $\(\Delta\)$ is set to 1 to amplify the differences between paired responses. This loss encourages the reward model (RM) to rank answers correctly by maximizing the margin between good and bad responses.
>
> **2.** **Hyperparameter** **Tuning**
>
> To ensure comparability, we employ the same hyperparameters for all training setups, including Loop 0, SER, and the full dataset. These include learning rate, batch size, and LoRA-specific hyperparameters, as detailed in Appendix C.3.
>
> The thresholds $\(\tau_{\text{high}}, \tau_{\text{low}}, \tau_{\Delta}\)$ were tuned extensively  in the self-labeling process. Specifically, we experimented with:
>
> - $\(\tau_{\text{high}} \in \{0.55, 0.65, 0.75\}\)$
> - $\(\tau_{\text{low}} \in \{0.45, 0.35, 0.25\}\)$
> - $\(\tau_{\Delta} \in \{0.3, 0.4, 0.5\}\)$
>
> Based on these experiments, we finalized:
>
> - $\(\tau_{\text{high}} = 0.55\)$
> - $\(\tau_{\text{low}} = 0.45\)$
> - $\(\tau_{\Delta} = 0.3\)$
>
> These values provided the most consistent improvements in RM performance while minimizing error amplification.
>
> 3. **Differences in Training Datasets**
>
> The primary difference between Loop 0, SER, and the full dataset lies in the training datasets:
>
> - **Loop 0:** Trained on 15% of the human-labeled data.
> - **Full Dataset:** Trained on the complete 100% human-labeled data.
> - **SER**: Combines 15% of the human-labeled data with self-labeled data filtered using the RM’s predictions.
>
> By keeping the loss function and hyperparameters consistent across all setups, we ensure a fair comparison of results. For further details, we refer the reviewer to Appendix C.3, which comprehensively outlines our training procedure.
>
> **Comment 2：**
>
> On eval: Have the authors tested on common benchmarks like RewardBench (so we can easily compare to other scalar and generative RMs)?
>
> Thank you for your comment. To evaluate the performance of our method, we tested our model, SER (trained with 15% human-labeled reward data + self-labeled reward data), alongside Loop 0 (trained with 15% human-labeled reward data) and the full dataset baseline (trained with 100% human-labeled reward data). All three models were trained on HH-RLHF and tested on RewardBench, a common benchmark for reward models. The chat score results are summarized below:
>
> | Model        | Llama 3 8B | Mistral 7B | Llama 2 13B |
> | ------------ | ---------- | ---------- | ----------- |
> | Loop 0       | 63.6       | 49.2       | 51.9        |
> | SER          | 85.7       | 75.1       | 80.2        |
> | Full Dataset | 86         | 80.7       | 84.9        |
>
> Our results demonstrate that SER significantly improves performance compared to Loop 0, showcasing the efficacy of self-labeled data in enhancing reward models. Notably, even on Llama 3 8B, SER achieves performance comparable to the full dataset baseline, which aligns with the experimental findings in our paper.

---

> ### Author Response · Authors · 2024-11-23
> **Response to Comment3**
>
> **Comment 3:**
>
> In real LLM training, the same dataset would always contain examples of multiple domains. In the same dataset, lots of examples could be on simple helpfulness prompts, and lots of examples are on coding/reasoning prompts. In this case, how would thresholding work? Would the model first learn simple prompts and then coding/reasoning/complicated prompts? Or would the model give up on coding/reasoning/complicated prompts?
>
> **Response to Comment 3:**
>
> Thank you for raising this point. We approach the problem as a general issue regarding the relationship between **easy** and **hard** tasks rather than focusing on task-specific domains (e.g., coding, mathematics, reasoning).
>
> - **Easy Tasks (Status 1):** These are tasks where the model can confidently distinguish between responses of differing quality, such as simple helpfulness prompts.
> - **Hard Tasks (Status 2):** These are tasks where the model struggles to differentiate between responses due to subtle distinctions (e.g., both responses are good or bad), requiring the model to amplify fine-grained differences.
>
> Our method is designed to address this general categorization of tasks, allowing the model to prioritize learning simple tasks (Status 1) first, building confidence, before progressing to harder tasks (Status 2). The thresholding mechanism ensures that both types of tasks are effectively addressed in a curriculum-learning-inspired framework.
>
> This universal approach to reward learning aligns with how other works in Reinforcement Learning from Human Feedback (RLHF)—such as PPO, DPO, SimPO, and self-rewarding methods—treat datasets. These methods do not separate tasks based on domain-specific properties (e.g., coding, reasoning) but rather optimize the model on the dataset as a whole, using a unified learning objective.
>
> In summary, our method ensures that thresholding works as intended across a mixed-domain dataset by modeling task difficulty (easy vs. hard) instead of specific domain characteristics. This makes the approach broadly applicable to real-world LLM training scenarios without the risk of "giving up" on more complex prompts like coding or reasoning.

---

> ### Author Response · Authors · 2024-11-23
> **Response to Comment4**
>
> **Comment 4：** Eq. (3) can be clearer:
>
> Thank you for your thoughtful questions. We appreciate the opportunity to clarify the status determination process and address any confusion caused by the notation in our paper.
>
> **Correcting Notation**
>
> Firstly, we apologize for the error in our notation regarding $\( \Delta_p \)$. In the paper, we incorrectly defined:
>
> $
> \Delta_p = \frac{1}{N} \sum_{i=1}^N |p_i^1 - p_i^2|.
> $
>
> The correct formulation should be:
>
> $
> \Delta_i = |p_i^1 - p_i^2|,
> $
>
> where $\( \Delta_i \)$ represents the absolute difference between the predicted probabilities for a pair of answers $\(A_i^1, A_i^2\)$ to the same question $\( Q_i \)$.
>
> **Status Determination**
>
> Our method employs two statuses to guide the reward model (RM) through a curriculum learning approach:
>
> **Status 1 (Easier Task):**
>
> Evaluates whether the RM can distinguish between positive (good) and negative (bad) samples.
>
> **Criteria:**
>
> - For an answer $\( A_{jk} \)$, the RM predicts a probability $\( p_{jk} \)$ of being good.
>   - **Positive Confidence:** If $\( p_{jk} > \tau_{\text{high}} \)$, the RM is confident the answer is good.
>   - **Negative Confidence:** If $\( p_{jk} < \tau_{\text{low}} \)$, the RM is confident the answer is bad.
> - **Status Achievement:** If the RM produces a sufficient number of high-confidence predictions (e.g., at least 600 samples in the HH dataset), we consider Status 1 achieved.
>
> **Status 2 (Harder Task):**
>
> Assesses whether the RM can discern subtle differences between answers of similar quality.
>
> **Criteria:**
>
> - For each pair of answers $\(A_i^1, A_i^2\)$ to the same question, compute $\( \Delta_i \)$:
> $
> \Delta_i = |p_i^1 - p_i^2|
> $
> - If $\( \Delta_i > \tau_\Delta \)$, the RM effectively highlights distinctions between similar-quality answers.
> - **Status Achievement:** If enough pairs meet this threshold (e.g., at least 600 pairs in the HH dataset), we consider Status 2 achieved.
>
> Sequential Status Checking
>
> - **Status 1 First:** We prioritize Status 1 because it represents a fundamental capability. If the RM satisfies Status 1, we proceed to the next iteration using data filtered based on this status.
> - **Status 2 Next:** If the RM does not meet the criteria for Status 1, we check for Status 2. If Status 2 is achieved, we proceed using data filtered based on Status 2.
> - **Stopping Criterion:** If the RM fails to meet the criteria for both statuses, we conclude that it has reached convergence and stop the iterative process.
>
> **Addressing Your Specific Questions**
>
> **Can Status 1 and Status 2 Be Achieved at the Same Time?**
>
> While both statuses could theoretically be satisfied simultaneously, we evaluate them sequentially to align with our curriculum learning approach. We first check for Status 1; only if it is not achieved do we check for Status 2. This sequential method ensures the RM builds foundational skills before tackling more complex tasks.
>
> **Does $ S = \text{Status1 or Status2} $?**
>
> - $\( S \)$ represents the current status of the RM in a given iteration:
>   - If the RM meets the criteria for Status 1, we set $\( S = \text{Status1} \)$.
>   - If the RM does not meet Status 1 but meets the criteria for Status 2, we set $\( S = \text{Status2} \)$.
>   - If the RM meets neither criterion, we set $\( S = \text{Stop} \)$, indicating convergence.
>
> **Do We Have a Different $\( S \)$ at Every i?**
>
> No, we do **not** have a different S at every i. The status S represents the overall state of the reward model (RM) during each training **iteration**, not per individual sample i.
>
> **Do We Need to Recompute $\( \Delta_i \)$ After Each Gradient Step?**
>
> No, we compute $\( \Delta_i = |p_i^1 - p_i^2| \)$ during the status checking phase after each training iteration, not after each gradient step. This computation helps determine whether the RM meets the criteria for Status 2.
>
> **When Would the Model Hit “Stop”?**
>
> The model reaches “Stop” when it fails to meet the criteria for both Status 1 and Status 2 after an iteration. This indicates that further self-training is unlikely to yield significant improvements, and we halt the iterative process.
>
> **What If the “Otherwise” Case Never Happens?**
>
> In practice, due to limitations in data and model capacity, the RM will eventually fail to meet the criteria for both statuses, leading to convergence. To ensure termination, we also set a maximum number of iterations as an additional stopping criterion.

---

> > ### Author Response · Authors · 2024-11-23
> > **Supplement regarding the 'response to Comment 4'**
> >
> > An explanation in natural language of SER.
> >
> >
> > **Iterative Loop Process**
> >
> > **Current Iteration**
> >
> > **Training Phase:**
> >
> > - Train the RM on the filtered data for a fixed number of epochs (2 epochs). This was determined through hyperparameter tuning and applied consistently for simplicity (see Appendix C).
> >
> > **Status Checking and Data Filtering:**
> >
> > 1. **Self-Labeling:**
> >    1. Use the updated RM to predict probabilities $\( p_i^1 \)$ and $\( p_i^2 \)$ for pairs of answers in the unlabeled data.
> > 2. **Status Checking:**
> >    1. Calculate $\( \Delta_i = |p_i^1 - p_i^2| \)$ for each pair.
> >    2. Determine the status $\( S \)$ based on the criteria:
> >       - If Status 1 is met (sufficient high-confidence predictions), set $\( S = \text{Status1} \)$.
> >       - Else, if Status 2 is met (sufficient pairs where $\( \Delta_i > \tau_\Delta \))$, set $\( S = \text{Status2} \)$.
> >       - Else, set $\( S = \text{Stop} \)$.
> > 3. **Data Filtering:**
> >    1. If $\( S = \text{Status1} \)$ or \$( S = \text{Status2} \)$, filter the data accordingly for the next iteration.
> >    2. If $\( S = \text{Stop} \)$, end the iterative process.
> >
> > **Transition to Next Iteration**
> >
> > 1. **Preparation:**
> >    1. The filtered dataset becomes the training set for the next iteration.
> >    2. Exclude data used in previous iterations to avoid redundancy.
> >
> > 2. **Next Iteration:**
> >    1. Repeat the training phase and status checking with the newly filtered data.

---

> ### Author Response · Authors · 2024-11-23
> **Response to Comment5 - Comment 7**
>
> **Comment 5:** Details are lacking on data generation/filtering
>
> **Response to Comment 5:**
>
> - (1)How specifically are responses generated?
>
> We utilized response pairs from existing datasets for training. It is important to highlight that the **labels in the** **training data** **of RM refer to the reward signals, not the responses themselves**.
>
> - In the SER setting, only 15% of the labeled data (reward signals) was used, while the remaining labels were removed, and a self-labeling approach was employed to generate training labels (reward signals).
> - The methods for generating response pairs varied across datasets:
>   - **Summarize Dataset:** All responses were generated by humans.
>   - **UltraFeedback Dataset:** Responses were produced by different base models.
>
> By focusing on learning reward signals rather than responses, our method leverages diverse sources of responses while aligning with the reward model's purpose.
>
> - (2)What percentage of data are filtered at different point in training?
>
> **Figure 3** illustrates the percentage of training data filtered from each loop, represented with different colors. The horizontal axis indicates the percentage of the training data relative to the full dataset. To clarify this further, we will include specific numerical values in Figure 3.
>
> For example, consider the LLama3 8b in HH-RLHF dataset as an example, the data composition for each loop:
>
> | Loop         | Data composition                                    | Data Num |
> | ------------ | --------------------------------------------------- | -------- |
> | Loop 0       | 15% human label data                                | 4273     |
> | Loop 1       | 15% human label data,loop1 selected self-label data | 9022     |
> | Loop 2       | Loop1 data, Loop2 selected self-label data          | 14241    |
> | Loop 3       | Loop2 data, Loop3 selected self-label data          | 15623    |
> | Full dataset | 100% human label data                               | 28492    |
>
> This breakdown illustrates how training data (reward signals) expands iteratively with self-labeled data, guided by the RM’s predictions.
>
> - (3) Are there more conclusive evidence that alternating between status are beneficial?
>
> Yes, our experiments provide strong evidence supporting the benefits of alternating between Status 1 and Status 2. For instance, on the HH-RLHF dataset with LLaMA 3 8B (**Figure 2 and Section 4.1.2**):
>
> - The RM’s performance improved from **56.9** to **64.9** during learning in Status 1.
> - Further learning in Status 2 increased performance from **64.9** to **68.56**.
>
> This demonstrates the advantage of transitioning between statuses to progressively refine the RM’s capabilities, starting from simpler tasks and advancing to more challenging ones.
>
> - (4) How are hyperparameters tuned -- based on what metrics?
>
> The detailed experimental setup is provided in **Appendix C (lines 883–900).** To ensure comparability, we used the same hyperparameters for all training setups, including Loop 0, SER, and the full dataset. These include learning rate, batch size, and LoRA-specific hyperparameters.
>
> For self-labeling, the thresholds $\(\tau_{\text{high}}, \tau_{\text{low}}, \tau_{\Delta}\)$ were tuned extensively on the accuracy in the self-labeling process. Specifically, we experimented with:
>
> - $\(\tau_{\text{high}} \in \{0.55, 0.65, 0.75\}\)$
> - $\(\tau_{\text{low}} \in \{0.45, 0.35, 0.25\}\)$
> - $\(\tau_{\Delta} \in \{0.3, 0.4, 0.5\}\)$
>
> Based on these experiments, we finalized:
>
> - $\(\tau_{\text{high}} = 0.55\)$
> - $\(\tau_{\text{low}} = 0.45\)$
> - $\(\tau_{\Delta} = 0.3\)$
>
> These values provided the most consistent improvements in RM performance while minimizing error amplification.
>
> **Comment 6: Other misc issues**
>
> **Response to Comment 6:** Thank you for your feedback and suggestions. We will carefully re-examine this issue to ensure that it is appropriately addressed in our revisions.
>
> **Comment 7: Are Llama models base models or instruction-tuned models?**
>
> **Response to Comment 7:** As mentioned in Section 4 Experiment and Appendix C, We followed a standard experimental procedure consistent with previous works, such as those on PPO, DPO, and similar methodologies. Specifically, we selected **base models** (Llama models) and performed supervised fine-tuning (SFT) on the specified dataset. This was followed by PPO training to refine the reward model. In Appendix C.3, we have provided a detailed of the training parameters.

---

> ### Author Response · Authors · 2024-11-25
> **Thanks for comments**
>
> Thank you for your insightful feedback on our manuscript. We have provided detailed responses to your comments and are happy to clarify any further questions you might have.

---

> ### Author Response · Authors · 2024-11-28
> **Response to 'still concerned about paragraph 3'**
>
> **Comment 8: still concerned about paragraph 3**
>
> Thank you for your thoughtful question and for taking the time to re-evaluate our work. We would like to clarify that in our method, the determination of **easy** or **hard** tasks is not based on human subjective judgment (e.g., coding being inherently hard or helpfulness being easy). Instead, it reflects the reward model’s own ability to learn and predict reward signals dynamically. This ensures the method is task-agnostic and generalizes well across multiple domains. Thus, thresholding is based on the reward model's ability to predict the quality of answers rather than being domain-specific.
>
> Our reward model is fine-tuned from foundational models, such as **Mistral (7B)**, **LLaMA 3 (8B)**  which already exhibit strong multi-domain generalization capabilities. By leveraging these models, our fine-tuned reward model does not “give up” on harder tasks like coding, reasoning, or math. Instead, it progressively learns to handle these tasks as it gains confidence through the SER process.
>
> To validate this, we present the **MT-Bench results for Mistral (7B)** on **coding**, **reasoning**, **math**, and overall performance:
>
> | Catagory  | Comparison          | Win         | Tie          | Lose        |
> | --------- | ------------------- | ----------- | ------------ | ----------- |
> | overall   | SER vs Full Dataset | 61（19.0%） | 214（66.9%） | 45（14.1%） |
> |           | SER vs SFT          | 73（22.8%） | 199（62.2%） | 48（15%）   |
> | Reasoning | SER vs Full Dataset | 13（32.5%） | 24（60%）    | 3（7.5%）   |
> |           | SER vs SFT          | 14（35%）   | 23（57.5%）  | 3（7.5%）   |
> | math      | SER vs Full Dataset | 6（15%）    | 34（85%）    | 0（0%）     |
> |           | SER vs SFT          | 5（12.5%）  | 34（85%）    | 1（2.5%）   |
> | coding    | SER vs Full Dataset | 5（12.5%）  | 30（75%）    | 5（12.5%）  |
> |           | SER vs SFT          | 6（15%）    | 30（75%）    | 4（10%）    |
>
> The results demonstrate that **SER** **achieves comparable performance with the full dataset** while significantly outperforming the baseline **Loop 0**. Notably, in **reasoning**, SER wins 35% of the time against SFT and only loses 7.5%, demonstrating its ability to handle complex tasks. Furthermore, the **overall results** show that SER maintains competitive performance across all categories, with 66.9% ties and only 14.1% losses against the full dataset.
>
> In summary, SER ensures robust multi-domain learning by dynamically adapting to the reward model’s predictive abilities. It does not “give up” on complex tasks like coding, reasoning, or math, but instead progresses through them effectively. Additionally, the **overall performance results** confirm that SER works well across diverse tasks, maintaining a high level of generalization. We hope this clarifies how thresholding works in our method and addresses your concerns. Thank you for your feedback.

---

> ### Author Response · Authors · 2024-11-28
> **Response to 'hopefully more detailed result on paragraph 2'**
>
> **Comment 9: hopefully more detailed result on paragraph 2**
>
> As stated in our previous response, we reported the **Reward Bench** evaluation results for the "HH-RLHF" reward model, which was trained on the helpful subset of the dataset. Specifically, we provided scores for the **chat component**, reflecting the model’s performance in alignment with its training focus.
>
> ### **Reward Model Performance**
>
> To further address your concerns, we also present results for the **RewardBench** evaluation of models trained on the **UltraFeedback dataset**. This dataset offers a comprehensive benchmark, allowing us to evaluate generalization across diverse tasks:
>
> | **Model**       | **Loop**  | **Avg.** | **Chat** | **Chat-Hard** | **Safety** | **Reasoning** |
> | --------------- | --------- | -------- | -------- | ------------- | ---------- | ------------- |
> | **Llama 3 8B**  | Loop 0    | 59.1     | 70.7     | 44.1          | 52.2       | 69.7          |
> |                 | SER       | 72.3     | 97.2     | 58.8          | 67.8       | 65.4          |
> |                 | Full Data | 75.0     | 95.5     | 58.5          | 73.9       | 72.0          |
> | **Mistral 7B**  | Loop 0    | 56.3     | 55.9     | 51.3          | 54.9       | 63.2          |
> |                 | SER       | 67.0     | 93.2     | 55.5          | 64.5       | 55.0          |
> |                 | Full Data | 66.8     | 91.3     | 54.2          | 64.1       | 60.4          |
> | **Llama 2 13B** | Loop 0    | 56.3     | 82.7     | 45.2          | 60.4       | 37.0          |
> |                 | SER       | 72.2     | 94.7     | 56.4          | 66.0       | 71.8          |
> |                 | Full Data | 74.1     | 95.5     | 64.0          | 68.7       | 68.3          |
>
> The results demonstrate that **SER** significantly enhances model performance compared to the baseline **Loop 0**, achieving results close to those obtained with the full human-annotated dataset. In tasks related to dialogue (Chat and Chat-Hard), which are central to the **UltraFeedback dataset**, **SER** achieves performance nearly identical to models trained on the full dataset.
>
> ### **PPO** **Model Performance**
>
> To further evaluate the downstream performance of models trained with **SER**, we provide results from **MT-Bench** and **Arena-Hard**, comparing **SER** to both the **Full Dataset** and **SFT** baselines:
>
> #### **MT-Bench Results:**
>
> | **Model**      | **Comparison**      | **Win**      | **Tie**     | **Lose**    |
> | -------------- | ------------------- | ------------ | ----------- | ----------- |
> | **Llama 3 8B** | SER vs Full Dataset | 96 (30.0%)   | 110 (34.3%) | 114 (35.6%) |
> |                | SER vs SFT          | 116 (36.25%) | 112 (35.0%) | 92 (28.8%)  |
> | **Mistral 7B** | SER vs Full Dataset | 61 (19.0%)   | 214 (66.9%) | 45 (14.1%)  |
> |                | SER vs SFT          | 73 (22.8%)   | 199 (62.2%) | 48 (15.0%)  |
>
> #### **Arena-Hard Results:**
>
> | **Model**      | **Comparison**      | **Win**    | **Tie**     | **Lose**   |
> | -------------- | ------------------- | ---------- | ----------- | ---------- |
> | **Llama 3 8B** | SER vs Full Dataset | 62 (12.4%) | 363 (72.6%) | 75 (15.0%) |
> |                | SER vs SFT          | 70 (14.0%) | 379 (75.8%) | 51 (10.2%) |
> | **Mistral 7B** | SER vs Full Dataset | 34 (6.8%)  | 442 (88.4%) | 24 (4.8%)  |
> |                | SER vs SFT          | 41 (8.2%)  | 435 (87.0%) | 24 (4.8%)  |
>
> The results highlight that **SER** achieves competitive performance relative to models trained on the full dataset and often surpasses SFT-trained models. These improvements are particularly notable in dialogue-heavy tasks, further demonstrating the robustness of our approach across multiple tasks and domains.
>
> In summary, our method achieves generalization by focusing on the reward model’s inherent ability to learn and predict reward signals dynamically, ensuring that no domain is neglected. Additionally, we have included more detailed experimental results as requested, further validating the generalization and effectiveness of our approach on three additional benchmarks: MT-Bench, Arena-Hard, and Reward-Bench. We sincerely hope this explanation clarifies our methodology and resolves any remaining concerns, highlighting the work's readiness **for an acceptable evaluation**. Thank you again for your thoughtful feedback and valuable insights.

---

> ### Author Response · Authors · 2024-12-02
> **Response to 'still concerned'**
>
> We truly value your thoughtful feedback and suggestions.  As the discussion period comes to a close, we hope our detailed responses have addressed your concerns and demonstrated the improvements made to strengthen the paper. We hope the improvements address your concerns and align with the criteria for an acceptable-level score.

---

### Official Review · Reviewer_U5TR · 2024-10-30

**Soundness:** 3
**Presentation:** 2
**Contribution:** 3
**Rating:** 6
**Confidence:** 4

**Summary:**

The paper proposed a self-improvement RM training strategy to circumvent the challenges of collecting human preference data. The RM is initially warmed up using only 15% human preference, then it’s improved iteratively through a series of self-labeling and retraining. During the self-labeling the current RM is used to score unlabeled data (i.e., prompt and answer pairs), then two data filtering strategies are devised based on the predicted probability differences between answer pairs. The goal is for the RM to (1) first learn from pairs with clear distinction (trained on very high- and very low-scored response pairs) before (2) switching to refining its understanding of subtle differences (trained on pairs that are more similar). When no improvement is seen on this current data, training stops and new data is collected.

The author tested their approach on multiple datasets including HH-RLHF, UltraFeedback as well as several model sizes and families including Llama and Mistral. They also evaluate how the improved RM helps during PPO.

**Strengths:**

- Promising results on improving the RM performance matching or exceeding the baseline train on the full human labeled data
- The curriculum learning style for the RM which encourage learning from easy and then hard pairs seems an interesting finding in the context of RM optimization

**Weaknesses:**

- The Experimental design is very narrow, and limited. (see questions for more details). E.g, I would expect to see results on downstream/standard benchmarks. However currently only preference accuracy is reported.
- Some claims in the results section are unsupported (mixed results which are generalized) → see detail below
- The paper clearly benefits from some rewriting and reorganization. In its current form several details are missing which makes it uneasy to follow.

**Questions:**

1. Are the pairs of responses generated from the same base LLM? Or are the authors using the response pairs in the existing dataset. While the author mentioned generating two responses from the LLM (line 173), it is not clear which model they use to generate responses and if they do this on-policy data generation for all the iterations? More details on this is appreciated.
2. Line 174-175: how generating two responses on-policy from the LLM "enriches" the training data with diverse answer qualities? What aspect of this setup is unique or does it guarantee quality/diversity?
3. Intuitively, what does the RM learn from its highly confident labeled data during stage 1? In other words, when the RM is too confident about a preference label (irrespective of whether it may be incorrect), what training signals it is gaining from it? Did authors try some ablations to test the contribution of each data state?
4. How the number of training steps differ in SER vs Full dataset in Table 1?
5. One would expect to see the performance of your PPO'ed model on downstream standard benchmarks like AlpacaEval, MMLU, BBH, MATH, CODEX, etc. So technically, start from a pre-trained base model doing supervised finetuning on some existing instruction-tuning data , then try doing PPO using your RM vs the baseline RM (trained on full data, e.g. on UltraFeedback) and compare performance over standard benchmarks. This will more realistically show the effectiveness of your approach as opposed to just narrowly evaluating models in their ability to choose the preferred response.
6. Line 343: The scaling claims are somewhat unsupported, based on Table 1 13b> 7b> 8b. more experiments specifically within the same model family is needed to make informative scaling trend conclusions.
7. Section 4.1.2, there's no explanation as to what Loop 1-3 refers to in section 4.1.2 and figure 2. It is not clear if loop is referring to one iteration of RM training (including both state 1 and 2), or something else? Line 373 states that "loop 2 is the least significant across all iterations" which makes me think loop != iteration
8. Line 418: does that mean Loop 1 and Loop 2 did not include data strategy of status 2? In general more details on the training is nice in the main paper.
9. Section 4.2: it would make sense to try PPO on a more general purpose preference dataset like ultrafeedback and see how it transfers to downstream tasks and not just alpacaeval style evaluation
10. Suggestion: the authors refer to their models as SER, or self RLHF. It’s nice to stick to one name and be consistent with it.

---

> ### Author Response · Authors · 2024-11-22
> **Response to Comment1 and  Comment2**
>
> **Comment 1**: Are the pairs of responses generated from the same base LLM? Or are the authors using the response pairs in the existing dataset. While the author mentioned generating two responses from the LLM (line 173), it is not clear which model they use to generate responses and if they do this on-policy data generation for all the iterations? More details on this is appreciated.
>
> **Response to Comment 1:** Thank you for your question. We clarify below how response pairs were generated and used in our experiments:
>
> 1. **Utilization of Existing Datasets:**
>    1. We do not generate two responses with LLMs, instead we use response pairs from existing datasets for training. We revised the unclear statement in lines 174-175.
>    2. In the SER setting, only 15% of the labeled data (including reward signals) were employed, and we removed the labels of the remaining 85% data while keeping the questions and their responses as unlabeled data. A self-labeling approach was then used to generate labels (reward signals instead of responses).
> 2. **Variation Across Datasets:**
>    1. The methods for generating response pairs varied across datasets:
>       - **Summarize Dataset:** All responses were generated by humans.
>       - **UltraFeedback Dataset:** Responses were produced by GPT-4.
>       - **HH-RLHF Dataset:** Responses were generated by different base models.
>       - **StackOverflow** **Dataset:** Human-generated responses were used.
> 3. **Clarity on Iterative Training:**
>    1. While the response pairs in existing datasets formed the basis of training, we did not perform on-policy response generation in any loop. Instead, we leveraged the self-labeling mechanism during iterative training to refine the reward model further.
> 4. **Addition to Table 2:**
>    1. To enhance clarity, we have added a column to Table 2 summarizing the response generation methods for each dataset.
>
> | Dataset       | Num      | Task      | Feedback Type | Response Type  |
> | ------------- | -------- | --------- | ------------- | -------------- |
> | Stackoverflow | 31284837 | QA        | human         | human response |
> | HH-RLHF       | 169352   | QA        | human         | LLM response   |
> | UltraFeedback | 63967    | QA        | GPT4          | LLM response   |
> | Summarize     | 179000   | Summarize | human         | human response |
>
> **Comment 2：** Line 174-175: how generating two responses on-policy from the LLM "enriches" the training data with diverse answer qualities? What aspect of this setup is unique or does it guarantee quality/diversity?
>
> **Response to comment 2:** Thank you for highlighting this point. We recognize that the original statement lacked clarity. To address this, we have revised lines 174–175 as follows:
>
> *"Furthermore, by allowing the reward model (RM) to judge two answers for each question, the RM is provided with paired examples that are essential for learning both statuses. This setup enables the RM to improve its discrimination and comparative abilities."*
>
> This setup contributes to quality and diversity in the training data for the following reasons:
>
> - By presenting two responses with varying qualities for the same question, the RM is exposed to both **positive vs. negative distinctions** (Status 1) and **nuanced differences** (Status 2). This enhances the RM's ability to handle diverse answer qualities effectively.

---

> ### Author Response · Authors · 2024-11-22
> **Response to Comment3 and Comment4**
>
> **Comment 3:** Intuitively, what does the RM learn from its highly confident labeled data during stage 1? In other words, when the RM is too confident about a preference label (irrespective of whether it may be incorrect), what training signals it is gaining from it? Did authors try some ablations to test the contribution of each data state?
>
> **Response to Comment 3:** Thank you for your insightful questions. We appreciate the opportunity to clarify what the reward model (RM) learns during **Status 1** and the contributions of each data status to its performance.
>
> **Learning from High-Confidence Data in Status 1:**
>
> During **Status 1**, the RM focuses on self-labeled data where it is highly confident about its predictions—specifically, samples where the predicted probabilities are significantly high (positive) or low (negative). Training on these high-confidence samples allows the RM to: (1) **Reinforce Fundamental Distinctions:** By concentrating on samples it confidently classifies, the RM strengthens its understanding of the key features that differentiate good responses from bad ones. This reinforcement helps solidify its foundational classification capabilities. (2） **Establish Clear Decision Boundaries:** High-confidence samples enable the RM to define more precise decision boundaries between classes, improving overall classification accuracy. （3) **Reduce Noise in** **Training Data****:** Filtering for high-confidence predictions minimizes the inclusion of ambiguous or noisy data, which could otherwise hinder learning and lead to overfitting or error propagation.
>
> **Addressing Potential Overconfidence and Incorrect Labels:**
>
> We acknowledge that high confidence does not guarantee correctness, especially in self-labeled data. However, the RM begins with initial training on human-labeled data, providing a reliable foundation. This setup helps mitigate the risks associated with overconfidence in incorrect labels: (1) **Statistical Likelihood of Correctness:** High-confidence predictions are statistically more likely to be correct, given the RM's baseline accuracy from human-labeled data. (2) **Minimizing Error Amplification:** By focusing on predominantly correct, high-confidence samples, the RM reduces the chance of reinforcing incorrect patterns.
>
> **Ablation Studies on Data Status Contributions:**
>
> We conducted ablation studies to assess the impact of **Status 2** on the RM's performance. We present the performance variations for each loop in **Figure 2**, where loop 1 and loop 2 correspond to status 1, and loop 3 corresponds to status 2. Additionally, **in the appendix (Figure 6)**.
>
> - **Status 1 is Essential but Not Sufficient Alone:** Training the RM solely on data from **Status 1** led to initial significant improvements compared to the initial state (Loop 0). However, the RM's performance plateaued after the early loops, showing limited further gains. For specific details, please refer to the analysis in Section 4.1.2 (lines 482-489).
> - **Status 2 Enables Advanced Learning: (1) Enhanced Discrimination:** Incorporating **Status 2** data allowed the RM to learn from more challenging samples, where distinctions between responses are subtle. (2) **Continued Improvement:** With Status 2, the RM continued to improve, demonstrating enhanced capability to discern nuanced differences between similar-quality responses.For specific details, please refer to the analysis in Section 4.1.2 (lines 491-501).
>
> **Comment 4**: How the number of training steps differ in SER vs Full dataset in Table 1?
>
> **Response to Comment 4**:Thank you for your question. As shown in Figure 2, the horizontal axis represents the number of training steps. The SER approach requires significantly fewer steps than the Full dataset while achieving comparable performance.
>
> For example, with the LLaMA 3 8B model on the HH-RLHF dataset:
>
> - **Loop 0:** Trained on 15% human-labeled data (4,273 samples) in **2****6****0 steps**.
> - **Loop 1:** Included data from Loop 0 and filtered self-labeled data, requiring **5****5****0 steps**.
> - **Loops 2–3:** Added more filtered data, with the total steps reaching **954**.
>
> In contrast, training on the Full dataset (28,492 samples) required 1782 steps due to its size. The SER approach achieves similar performance using fewer steps and less human-labeled data, demonstrating its efficiency.

---

> ### Author Response · Authors · 2024-11-22
> **Response to Comment5 - Comment7**
>
> **Comment 5:** One would expect to see the performance of your PPO'ed model on downstream standard benchmarks like AlpacaEval, MMLU, BBH, MATH, CODEX, etc. So technically, start from a pre-trained base model doing supervised finetuning on some existing instruction-tuning data , then try doing PPO using your RM vs the baseline RM (trained on full data, e.g. on UltraFeedback) and compare performance over standard benchmarks. This will more realistically show the effectiveness of your approach as opposed to just narrowly evaluating models in their ability to choose the preferred response.
>
> **Response to Comment 5:** Thank you for the suggestion regarding the evaluation prompts. To address this, we have included results using the MT-Bench LLM-as-a-judge prompt, which has been shown to correlate well with human judgments. Below are the detailed results from MT-Bench evaluations:
>
> | Model          | Comparison          | Win           | Tie          | Lose         |
> | -------------- | ------------------- | ------------- | ------------ | ------------ |
> | **Llama 3 8B** | SER vs Full Dataset | 96（30%）     | 110（34.3%） | 114（35.6%） |
> |                | SER vs SFT          | 116（36.25%） | 112（35%）   | 92（28.8%）  |
> | **Mistral 7B** | SER vs Full Dataset | 61（19.0%）   | 214（66.9%） | 45（14.1%）  |
> |                | SER vs SFT          | 73（22.8%）   | 199（62.2%） | 48（15%）    |
>
> These results are consistent with those presented in the paper, demonstrating that SER achieves strong performance compared to both the full dataset and SFT baselines. In addition to MT-Bench, we are currently running evaluations using the Arena-Hard benchmark. We will update the results once the training and evaluation process is complete.
>
> **Comment 6:** Line 343: The scaling claims are somewhat unsupported, based on Table 1 13b> 7b> 8b. more experiments specifically within the same model family is needed to make informative scaling trend conclusions.
>
> **Response to Comment 6:** Thank you for your feedback. We recognize the need for more robust scaling experiments and have revised lines 343–346 as follows:
>
> *"A potential trend is observed where the difference between the* *SER* *method and the full human-labeled data increases with model size. Specifically, the average difference for Mistral 7B is +0.12%, for LLaMA 8B is +0.06%, and for LLaMA 13B is -1.07%. This suggests that larger models better utilize labeled data, enhancing performance. This trend highlights the potential of SER to further elevate model performance by scaling labeled data through self-labeling rather than manual annotation."*
>
> To address concerns regarding model families, we chose widely-used base models in alignment with prior studies. In future work, we aim to conduct scaling experiments within a single model family to strengthen the credibility and informativeness of our scaling claims.
>
> **Comment 7:** Section 4.1.2, there's no explanation as to what Loop 1-3 refers to in section 4.1.2 and figure 2. It is not clear if loop is referring to one iteration of RM training (including both state 1 and 2), or something else? Line 373 states that "loop 2 is the least significant across all iterations" which makes me think loop != iteration
>
> **Response to Comment 7:** Thank you for your question. In our paper, a "loop" is equivalent to an iteration, as shown in Figure 1. Each iteration (or loop) consists of the following sequence:
>
> 1. **Step 1:** Self-labeled data with the reward model (RM).
> 2. **Step 2:** Select high-confidence self-labeled data.
> 3. **Step 3:** Retrain the RM with the selected self-labeled data.
>
> The newly trained RM is then used in the next iteration (or loop) to continue the process.
>
> Regarding line 485, the statement "loop 2 is the least significant across all iterations" refers to our observation that, during the iterative process, the performance gain achieved in loop 2 was smaller compared to the other loops (iterations). We will revise the text in Section 4.1.2 to clarify that "loop" and "iteration" are used interchangeably, ensuring there is no ambiguity in interpretation.

---

> ### Author Response · Authors · 2024-11-22
> **Reponse to Comment8 - Comment10**
>
> **Comment 8：** Line 418: does that mean Loop 1 and Loop 2 did not include data strategy of status 2? In general, more details on the training is nice in the main paper.
>
> **Response to Comment 8:** Yes, that's correct. Loop 1 and Loop 2 did not include the data strategy of Status 2. During these loops, the reward model (RM) focused exclusively on Status 1, which involves distinguishing between positive (good) and negative (bad) samples based on high-confidence predictions. This approach allowed the RM to build a solid foundation in basic classification tasks.
>
> **Status Determination and Training Process**
>
> - Status 1 (Easier Task)
>   - **Objective:**     Evaluate whether the RM can clearly distinguish between positive and negative samples.
>   - **Criteria:**     For each answer $\( A_{jk} \)$:     - If $\( p_{jk} > \tau_{\text{high}} \)$, the RM is confident the answer is positive.     - If $\( p_{jk} < \tau_{\text{low}} \)$, the RM is confident the answer is negative.
>   - **Selection:**     Status 1 is chosen when there are enough high-confidence predictions (e.g., at least 600 samples in the HH dataset).
> - **Status 2 (Harder Task)**
>   - **Objective:**     Assess the RM's ability to discern subtle differences between answers of similar quality.
>   - **Criteria:**     For paired answers to the same question, compute the absolute difference:     $\(   |p_{j1} - p_{j2}| > \tau_\Delta   \)$
>   - **Selection:**     Status 2 is applied when enough pairs meet this threshold (e.g., at least 600 pairs).
>
> **Training Progression**
>
> - Loops 1 and 2: Focused on Status 1 to ensure the RM could confidently classify positive and negative samples. - Allowed the RM to learn from easier tasks before progressing to more complex ones.
> - Subsequent Loops: Introduced Status 2 after the RM demonstrated proficiency in Status 1. - Enabled the RM to tackle harder tasks, such as distinguishing between similar-quality answers.
>
> We agree that more details on the training process would enhance the clarity of our paper. In the revised manuscript, we have done the below changes:   1. Include a concise description of the training procedure in the main text.   2. Explain the roles of Status 1 and Status 2 and how the RM transitions between them.   3. Detail the loop iterations to provide a better understanding of how the RM evolves through the training stages.
>
> **Comment 9：** Section 4.2: it would make sense to try PPO on a more general purpose preference dataset like **ultrafeedback** and see how it transfers to downstream tasks and not just alpacaeval style evaluation.
>
> **Response to Comment 9:** Thank you for your suggestion. We are currently running this experiment and will share the results once training is complete.
>
> **Comment 10：** Suggestion: the authors refer to their models as SER, or self-RLHF. It’s nice to stick to one name and be consistent with it.
>
> **Response to Comment 10:** Thank you for your suggestion. We recognize the importance of consistency in naming conventions to avoid confusion. In the revised manuscript, we will ensure consistent usage of a single name throughout the paper. Specifically, we will refer to our method as SER for clarity and uniformity.

---

> > ### Comment · Reviewer_U5TR · 2024-11-24
> > **Thanks for response**
> >
> > I appreciate the author’s detailed responses, which addressed and clarified some of my concerns.
> >
> > While I appreciate the amount of work that has been done (various datasets and model sizes), as the author may have realized, the paper lacks enough clarity, structure, key methodological and experimental details—issues also noted by several other reviewers. Furthermore, more robust comparisons and experiments are necessary to effectively evaluate the efficacy and generalization of the proposed self-improving model.
> >
> > For now I will keep my score as is. Will finalize it after discussion with other reviewers.
> >
> > Thanks

---

> > > ### Author Response · Authors · 2024-11-26
> > > **Response to Comment11**
> > >
> > > **Comment11：**
> > > as the author may have realized, the paper lacks enough clarity, structure, key methodological and experimental details—issues also noted by several other reviewers. Furthermore, more robust comparisons and experiments are necessary.
> > >
> > >  **Response to Comment:**
> > >
> > > **（1）as the author may have realized, the paper lacks enough clarity, structure, key methodological and experimental details—issues also noted by several other reviewers.**
> > >
> > > Thank you for your feedback and for highlighting areas where our paper could be improved.   We genuinely appreciate your thoughtful comments, which have helped us refine and clarify our work.
> > >
> > > We understand your concern that the updates could suggest the submitted paper lacked details. However, we want to assure you that our intent was not to compensate for omissions but to further enhance clarity and address any potential ambiguity that might arise from the original structure. Scientific work often benefits from iterative improvements, and your insights have been invaluable in guiding this process. We believe that these enhancements, guided by your feedback, significantly improve the paper’s impact and accessibility without altering its core contributions.
> > >
> > > **（2） Furthermore, more robust comparisons and experiments are necessary to effectively evaluate the efficacy and generalization of the proposed self-improving model.**
> > >
> > > **Regarding the RM performance**, in addition to evaluating the RM's accuracy as outlined in the paper, we have also used the open-source benchmark, Reward Bench, for further evaluation. The results are presented below.
> > >
> > > To evaluate the performance of our method, we tested our model, SER (trained with 15% human-labeled reward data + self-labeled reward data), alongside Loop 0 (trained with 15% human-labeled reward data) and the full dataset baseline (trained with 100% human-labeled reward data). All three models were trained on HH-RLHF and tested on RewardBench, a common benchmark for reward models. The chat score results are summarized below:
> > >
> > > | Model        | Llama 3 8B | Mistral 7B | Llama 2 13B |
> > > | ------------ | ---------- | ---------- | ----------- |
> > > | Loop 0       | 63.6       | 49.2       | 51.9        |
> > > | SER          | 85.7       | 75.1       | 80.2        |
> > > | Full Dataset | 86         | 80.7       | 84.9        |
> > >
> > > Our results demonstrate that SER significantly improves performance compared to Loop 0, showcasing the efficacy of self-labeled data in enhancing reward models. Notably, even on Llama 3 8B, SER achieves performance comparable to the full dataset baseline, which aligns with the experimental findings in our paper.
> > >
> > > **Regarding the PPO model performance**, as per the reviewer's suggestion, we have included the test results on **MT-bench** and **Arena-hard**. The **MT-bench** results have already been provided in our response to comment 5, while the results for **Arena-hard** are presented below. To provide a clearer comparison of the relative performance differences between models, we have displayed the results in a win-tie-lose format.
> > >
> > > | Model          | Comparison          | Win       | Tie        | Lose      |
> > > | -------------- | ------------------- | --------- | ---------- | --------- |
> > > | **Llama 3 8B** | SER vs Full Dataset | 62(12.4%) | 363(72.6%) | 75(15%)   |
> > > |                | SER vs SFT          | 70(14%)   | 379(75.8%) | 51(10.2%) |
> > > | **Mistral 7B** | SER vs Full Dataset | 34(6.8%)  | 442(88.4%) | 24(4.8%)  |
> > > |                | SER vs SFT          | 41(8.2%)  | 435(87%)   | 24(4.8%)  |
> > >
> > > These results are consistent with those presented in the paper, demonstrating that SER achieves strong performance compared to both the full dataset and SFT baselines.
> > >
> > > Thank you for your insightful suggestion regarding additional benchmarks like MMLU, BBH, MATH, and CODEX. While these benchmarks are **not** commonly used during the RLHF stage or in the RLHF-related papers we referenced (e.g., PPO, DPO, SimPo, TDPO, self-rewarding, RLAIF, etc.), we greatly value your perspective. If you believe including these evaluations would enhance the paper, we would be more than happy to conduct the experiments and provide the results. We truly appreciate your thoughtful feedback and suggestions to further strengthen our work.
> > >
> > > **Regarding the comparisons**, we have provided results based on 15% LLM-labeled data and results using 60% human-labeled data. The results are shown below. For a detailed analysis, please refer to our response to reviewer 8P3z.
> > >
> > > | **Method**              | **Accuracy (%)** |
> > > | ----------------------- | ---------------- |
> > > | 60% Human-labeled +SER  | 71.83            |
> > > | Full Dataset            | 70.45            |
> > > | 15% Human-Labeled + SER | 68.56            |
> > > | 15% LLM-Labeled + SER   | 67.64            |
> > >
> > > We respectfully ask you to reconsider your score in light of these revisions and remain open to further suggestions. Thank you again for your valuable feedback and for helping us improve our work.

---

> > > > ### Comment · Reviewer_U5TR · 2024-11-28
> > > >
> > > > Thanks much for additional experiments on more open-source benchmarks as well as new results on applying SER on stronger RMs.
> > > >
> > > > I have increased my score!

---

> > > > > ### Author Response · Authors · 2024-11-28
> > > > > **Response to Reviewer U5TR**
> > > > >
> > > > > On this Thanksgiving Day, we want to express our heartfelt gratitude for your thoughtful feedback and for raising the score of our paper.

---

### Official Review · Reviewer_vMDv · 2024-11-01

**Soundness:** 3
**Presentation:** 2
**Contribution:** 2
**Rating:** 6
**Confidence:** 3

**Summary:**

This paper introduces an approach called Self-Evolved Reward Learning (SER), which leverages an RM to autonomously curate its training data for continuous improvement. The authors conduct experiments across various models and datasets, demonstrating that the SER approach can train the reward model well using only 15% of the labeled data, which results in performance comparable to an RM trained on the full labeled dataset, both in terms of evaluation capability and integration within PPO training.

**Strengths:**

**Motivation:** Given the high costs associated with acquiring human-annotated data for reward model training, the development of self-training approaches is highly beneficial. Such methods can leverage a minimal amount of human annotations for initial model training and subsequently enhance performance through self-improvement.

**Soundness:** The paper's various training strategies, which are based on different statuses, are well-conceived and intuitively sound.

**Substance:** The authors have conducted extensive experiments across multiple models and datasets, effectively demonstrating the efficacy of their Self-Evolved Reward Learning (SER) approach.

**Weaknesses:**

**Clarity:** The paper lacks clarity in several places, with some descriptions being particularly confusing. For instance:

- **Lines 206-211:** It remains unclear how the status is determined, given that $p_i^1$ and $p_i^2$ are both single numbers. How are these individual numbers used to decide the training status? Additionally, how are the thresholds $\tau_\text{high}$, $\tau_\text{low}$, and $\tau_\Delta$ established?

- **Lines 248-249:** The method for determining the $\Delta$ term is not explained.

- **Section 3.2 (Theoretical Analysis in Appendix A):** This section appears to offer only speculative insights into how the algorithm might function, rather than providing concrete proofs.

- **Lines 325-326:** The statement "Larger parameter models can achieve higher self-improvement benefits" seems inconsistent, as Llama 13B appears to exhibit lower benefits compared to Llama 8B and Mistral 7B.

- **Section 4.1.2 (Fine-grained Analysis):** The paper does not clearly define what Loop0, Loop1, Loop2, and Loop3 training entail. What data are these loops trained on, and do they use the same subset of data? How is the transition to the next loop decided? Furthermore, why is the strategy of learning status 2 employed in Loop 3 not earlier?

**Meaningful Comparisons:** In Figure 4, the comparison using GPT-4 as a judge seems to employ an unconventional prompt for evaluating model responses (Appendix D.4). Why not use the AlpacaEval or MT-bench LLM-as-a-judge prompts, which have been shown to correlate well with human judgments?

**Implications:** The paper currently focuses on improving reward models that are not well trained (i.e., trained on only a small subset of the complete human-annotated dataset). Ultimately, the goal is to continuously enhance strong reward models with minimal human annotation. It is unclear whether this approach can be applied to strong reward models and whether these models have the potential for continuous self-improvement. Exploring this line of experimentation would be valuable.

**Questions:**

See questions in the weaknesses part.

---

> ### Author Response · Authors · 2024-11-21
> **Reponse to Comment1 and Comment2**
>
> Dear reviewer,
>
> We thank you for the very helpful suggestions and we provide below with our answers to your comments.
>
> **Comment1**  :**Lines 206-211:** It remains unclear how the status is determined, given that pi1 and pi2 are both single numbers. How are these individual numbers used to decide the training status? Additionally, how are the thresholds τhigh, τlow, and τΔ established?
>
> **Response to Comment 1:**
>
> **Status Determination**:
>
> To determine the status, we use the reward model (RM) trained in the current loop to predict the probability of an answer being good on the **unlabeled data**. The two statuses are proposed to train RM from easy to hard samples (Line 189-225). To be more specific:
>
> \- **Status 1 (Easier Task):**     This status evaluates whether the RM can clearly distinguish between positive (good) and negative (bad) samples. The evaluation is based on the predicted probabilities $\(p_j^k\)$ for each answer $\(A_j^k\)$:
>
> \- If $\(p_j^k > \tau_{\text{high}}\)$, the RM is confident that the answer is positive.
>
> \- If $\(p_j^k < \tau_{\text{low}}\)$, the RM is confident that the answer is negative.
>
> Status 1 is selected when there is a sufficient number of these high-confidence predictions (e.g., 600 in the HH dataset), indicating that the RM is proficient in distinguishing positive and negative samples.
>
> \- **Status 2 (Harder Task):**     This status assesses the RM’s ability to discern subtle differences between answers of similar quality (e.g., both answers being good or both being bad). It requires the RM to evaluate paired answers to the same question and compute the absolute difference between their predicted probabilities:    $ \[   \left| p_j^1 - p_j^2 \right| > \tau_\Delta   \] $
>
> If enough paired predictions meet this threshold (e.g., 600 predictions in the HH dataset), it indicates that the RM can effectively highlight distinctions between similar-quality answers. This task is more challenging than Status 1 because it requires the RM to recognize and quantify nuanced differences.
>
> We check the statuses **in order**—first Status 1 and then Status 2. Status 1 must be met first as it represents a basic capability necessary before moving on to the more complex task in Status 2. If the RM does not meet the criteria for Status 1 (i.e., few or no samples satisfy the thresholds $\tau_{\text{high}}$ and $\tau_{\text{low}})$, we then check for Status 2. If the RM also fails to meet the criteria for Status 2, we interpret this as the model reached its convergence point, and we halt further training.
>
> **Establishment of Thresholds**:
>
> The thresholds $\tau_{\text{high}}$, $\tau_{\text{low}}$, and $\tau_\Delta$ were determined through extensive hyper-parameter tuning in the self-training process. Specifically, we experimented with the following values:
>
> - $\tau_{\text{high}} \in \{0.55, 0.65, 0.75\}$
> - $\tau_{\text{low}} \in \{0.45, 0.35, 0.25\}$
> - $\tau_\Delta \in \{0.3, 0.4, 0.5\}$
>
> After evaluating the RM's performance with these parameters, we selected $\tau_{\text{high}} = 0.55$, $\tau_{\text{low}} = 0.45$, and $\tau_\Delta$ = 0.3 as they provided the most consistent improvements in the RM's ability to self-label effectively without introducing significant error amplification.
>
> We have revised the manuscript (in blue color) to clarify how the statuses are determined, emphasizing that **Status 1 is the foundational (easier) task of distinguishing between positive and negative samples,** **while** **Status 2 is the more challenging task of discerning subtle differences between similar-quality answers**. This sequential approach reflects a curriculum learning strategy, progressing from simpler to more complex tasks.
>
> **Comment2** : **Lines 248-249:** The method for determining the Δ term is not explained.
>
> The \Delta term in our approach serves as a hyperparameter in the hinge loss function (set to 1 through search), designed to amplify the differences between two responses. For example, if the difference between two responses is small ($A_j^1$ is supposed to be better than $A_j^2$), the loss is larger. We drew inspiration from the hinge loss used in Support Vector Machines (SVM), where the margin parameter allows the model to maximize the separation between classes.

---

> ### Author Response · Authors · 2024-11-21
> **Response to Comment3 and Comment4**
>
> **Comment 3:** Section 3.2 (Theoretical Analysis in Appendix A): This section appears to offer only speculative insights into how the algorithm might function, rather than providing concrete proofs.
>
> **Response to Comment 3:**
>
> Thank you for your insightful feedback regarding the theoretical analysis section of our paper. In response to your comments, we have substantially revised this section to include formal statements, detailed assumptions, and rigorous proofs (highlighted with blue color in Appendix A).
>
> Specifically:
>
> 1. **Convergence** **of the Reward Model during Self-Training**: We have provided a formal theorem with clearly stated assumptions and a step-by-step proof demonstrating the conditions under which the reward model converges to a good solution through self-training.
> 2. **Convergence** **Properties of** **PPO** **with a Learned Reward Model**: We have included a comprehensive theoretical analysis supported by a theorem and proof that establish the convergence properties of PPO when using the improved reward model.
>
> **Comment 4**： '**Lines 325-326:** The statement "Larger parameter models can achieve higher self-improvement benefits" seems inconsistent, as Llama 13B appears to exhibit lower benefits compared to Llama 8B and Mistral 7B.**'**
>
> **Response to Comment 4:**
>
> Thank you for pointing it out. The original statement did not accurately reflect the findings. Based on the results in Table 1, we find that in 10 out of 12 comparisons between smaller and larger parameter models, the larger parameter models achieve higher self-improvement benefits. This supports the general trend that larger models typically gain more from self-improvement.
>
> To address this, the content in first paragraph Section 4.1.1 has been revised to: **"Larger parameter models typically achieve higher performance after undergoing self-improvement. In most experimental settings, the performance of the Llama 13B model surpasses that of the other two smaller parameter models."**
>
> This revision more effectively conveys the trends observed in our experimental results.

---

> ### Author Response · Authors · 2024-11-21
> **Response to Comment5**
>
> **Comment 5:** **Section 4.1.2 (Fine-grained Analysis):**
>
> **Response to Comment 5:**
>
> **The paper does not clearly define what Loop0, Loop1, Loop2, and Loop3 training entail. What data are these loops trained on, and do they use the same subset of data?**
>
> The descriptions of these loops are provided in Appendix C Reward Model Training (lines 839-840) , but we acknowledge that additional clarity can enhance the reader's understanding.
>
> To clarify:
>
> - **Loop 0**: The training of the reward model begins with 15% of the human-labeled data. This initial stage is referred to as Loop 0.
> - **Loops 1–3**: For subsequent loops, the training incorporates the data from the previous loop along with new data filtered through model self-labeling.
>
> We will utilize the following formal expressions to describe the variations in the training data as a supplementary explanation to Equation 5, should the reviewer consider it necessary:
>
> ${D_{\text{data}}}$ = $D^n_{\text{Self-labeled}}$ + $\{D^{n-1}_{\text{data}}}$
>
> here \(n\) denote the number of iterations of the loop. The training data for the current loop consists of the data filtered using Equation 4, in addition to the training data from the previous loops. We revised shown in lines 278-281. The newly added data for each loop, along with the total data, is presented in Figure 2.
>
> | Loop         | Data composition                                    | Data Num |
> | ------------ | --------------------------------------------------- | -------- |
> | Loop 0       | 15% human labeled data                                | 4273     |
> | Loop 1       | 15% human labeled data,loop1 selected self-labeled data | 9022     |
> | Loop 2       | Loop1 data, Loop2 selected self-labeled data          | 14241    |
> | Loop 3       | Loop2 data, Loop3 selected self-labeled data          | 15623    |
> | Full dataset | 100% human labeled data                               | 28492    |
>
> **How is the transition to the next loop decided?**
>
> The iterative loop operates as follows:
>
> - **Current Loop**:
>   - **Training Phase**: Train the reward model (RM) on the filtered data for a fixed number of epochs (**2 epochs**). We chose 2 epochs after hyperparameter tuning on the Stack Overflow dataset and applied this setting consistently across other datasets for simplicity (see Appendix C, lines 821-823).
>   - **Status Checking and Data Filtering**:
>     1. **Self-Labeling**: Use the updated RM from the current loop to perform self-labeling on the unlabeled data. The RM generates predictions, effectively labeling new data.
>     2. **Status Checking**: We evaluate the RM's status to determine if it meets the criteria for **Status 1** or **Status 2**. If the RM meets the criteria for **Status 1** or **Status 2**, we proceed to the next loop. **Stopping Criterion**: If the RM does **not** meet the criteria for either status, we interpret this as the model reaching convergence and **stopping** **the iterative process**.
>     3. **Data Filtering**: If proceeding, we use RM's predictions to select high-confidence data based on the status criteria.
> - **Transition to Next Loop**:
>   - **Preparation for Next Loop**: Apply the status criteria to the newly labeled data to select high-confidence samples. This filtered dataset becomes the training set for the next loop.
>
> This process repeats iteratively:
>
> - **Next Loop**:
>   - **Training Phase**: Train the RM on the filtered data (selected high-confidence data) from the previous loop.
>   - **Status Checking and Data Filtering**: Perform self-labeling on the unlabeled data (exclude the training data which is part of unlabeled data in previous loop), evaluate the RM's status and filter data for subsequent training.
>
> The transition to the next loop is determined by the RM's ability to meet the criteria for **Status 1** or **Status 2** after each training phase. If, at any point, the RM does not satisfy either status, we conclude that the RM has reached its convergence point and **stop the iterative process**.
>
> **Furthermore, why is the strategy of learning status 2 employed in Loop 3 not earlier?**
>
> We introduce the strategy for **Status 2** starting in **Loop 3** to align with a curriculum learning approach, as described in **lines 220-225**. Initially, we focus on **Status 1** (the easier task) to ensure the reward model (RM) can effectively distinguish between positive and negative samples. This establishes a solid foundation. Once the Status 1 criteria is not met (no enough selected data for training), it goes to **Status 2** (the harder task) of discerning subtle differences between similar-quality answers. Thus **status 2  is always employed** **at** **the end of the loop.**

---

> ### Author Response · Authors · 2024-11-21
> **Response to Comment6 and Comment7**
>
> **Comment 6**: **Meaningful Comparisons:** In Figure 4, the comparison using GPT-4 as a judge seems to employ an unconventional prompt for evaluating model responses (Appendix D.4). Why not use the AlpacaEval or MT-bench LLM-as-a-judge prompts, which have been shown to correlate well with human judgments?
>
> **Response to Comment 6**:
>
> Thank you for the suggestion regarding the evaluation prompts. To address this, we have included results using the MT-Bench LLM-as-a-judge prompt, which has been shown to correlate well with human judgments. Below are the detailed results from MT-Bench evaluations:
>
> | Model          | Comparison          | Win           | Tie          | Lose         |
> | -------------- | ------------------- | ------------- | ------------ | ------------ |
> | **Llama 3 8B** | SER vs Full Dataset | 96（30%）     | 110（34.3%） | 114（35.6%） |
> |                | SER vs SFT          | 116（36.25%） | 112（35%）   | 92（28.8%）  |
> | **Mistral 7B** | SER vs Full Dataset | 61（19.0%）   | 214（66.9%） | 45（14.1%）  |
> |                | SER vs SFT          | 73（22.8%）   | 199（62.2%） | 48（15%）    |
>
> These results are consistent with those presented in the paper, demonstrating that SER achieves strong performance compared to both the full dataset and SFT baselines. In addition to MT-Bench, we are currently running evaluations using the Arena-Hard benchmark. We will update the results once the training and evaluation process is complete.
>
> **Comment 7**: **Implications:** The paper currently focuses on improving reward models that are not well trained (i.e., trained on only a small subset of the complete human-annotated dataset). Ultimately, the goal is to continuously enhance strong reward models with minimal human annotation. It is unclear whether this approach can be applied to strong reward models and whether these models have the potential for continuous self-improvement. Exploring this line of experimentation would be valuable.
>
> To address the potential of applying our approach to strong reward models and enabling continuous self-improvement, we are conducting experiments with a strong reward model trained on 60% of the human-labeled data. We will share the results here after finishing the results.

---

> > ### Author Response · Authors · 2024-11-25
> > **Supplement regarding the 'response to Comment 7'**
> >
> > We utilized 60% human-labeled data(in HH-RLHF dataset), and the results are as follows. It can be observed that the use of SER further enhanced the performance of the RM, surpassing the performance achieved with the "full dataset". This validates the effectiveness of our method.
> >
> > | Method                 | ACC   |
> > | ---------------------- | ----- |
> > | SER(15% human-labeled) | 68.56 |
> > | SER(60% human-labeled) | 71.83 |
> > | Full dataset           | 70.45 |

---

> ### Author Response · Authors · 2024-11-25
> **Thanks for Comments**
>
> Thank you for your insightful feedback on our manuscript. We have provided detailed responses to your comments and are happy to clarify any further questions you might have.

---

> ### Comment · Reviewer_vMDv · 2024-11-25
> **Thanks for the author response**
>
> Thanks for the author response and I have increased my score.

---

> > ### Author Response · Authors · 2024-11-26
> > **Thanks for valuable feedback**
> >
> > Thank you for your thoughtful review and the increased scores! We are glad the revisions aligned with your expectations, and we appreciate your valuable feedback.

---

### Official Review · Reviewer_8P3z · 2024-11-10

**Soundness:** 3
**Presentation:** 4
**Contribution:** 3
**Rating:** 6
**Confidence:** 3

**Summary:**

The manuscript introduces an approach to improving sample-efficiency of RLHF. The method enables using only a fraction of the original data through an iterative approach of labelling and ranking of the preference data based on the dynamics of the training process. The authors provide theoretical support of their empirical results, as well as extensive experiments (multiple models, multiple datasets).

**Strengths:**

The manuscript, in my opinion, has several strengths that make it interesting to the ICLR audience.

* Tackles an important problem, namely of sample efficiency, for RLHF methods. The cost of human annotations is rather high and therefore studying ways to reduce the dependency on the entire datasets is useful overall.

* The recipe is well presented and supported theoretically. Although the steps seem obvious at first, the subtlety of amplifying the differences (second step) during the training loop seems rather novel and not extensively studied before. Effectively, it introduces the ability for the model to provide signal as to what samples provide most utility at current training stage.

* The extensive results, which utilize multiple models and multiple datasets. The performance across all these combinations of setups seems to support that the recipe is effective and general.

**Weaknesses:**

The main weakness in current practices is the longer-term importance of annotated human datasets. The recipe introduces significant compute costs, trading off annotation costs for accelerators (through the recipe steps). However, the manuscript does not study the relationship between the two. If we assume human-annotated data goes away for preference labelling - having datasets created by LLMs for training other LLMs - do the benefits of the recipe still outweigh the costs?

I would like to understand the authors' position on this aspect, as well as a better comparison between labeling costs and accelerator costs of their recipe. Where does a tie come in (e.g. at the K=4 loops used in the paper, or for what value of K)?

**Questions:**

Please refer to the weakness comment.

---

> ### Author Response · Authors · 2024-11-22
> **Response to Comment1**
>
> **Comment 1**： If we assume human-annotated data goes away for preference labelling - having datasets created by LLMs for training other LLMs - do the benefits of the recipe still outweigh the costs? Where does a tie come in (e.g. at the K=4 loops used in the paper, or for what value of K)?
>
> **Response to Comment1**:Thank you for raising this important point. Below, we provide a detailed comparison of the costs associated with human labeling, the SER method, and LLM-generated labels, alongside their respective performances. We also discuss the broader implications of the SER method.
>
> ## **Cost Comparison**
>
> ### **1. Human Labeling Costs**
>
> Human labeling involves reading a prompt and two candidate responses (average total: 304 words).
>
> - **Google Cloud Pricing:** $0.11 USD per 50 words for classification tasks.
> - **Cost per Example:**$\(\text{Cost} = \frac{304 \, \text{words} \times 0.11 \, \text{USD}}{50 \, \text{words}} = 0.67 \, \text{USD/sample}\)$
>
> | **Method**          | **Human-Labeled Data Used** | **Cost per Sample (****USD****)** |
> | ------------------- | --------------------------- | --------------------------------- |
> | Full Human Labeling | 100%                        | 0.67                              |
> | SER Method          | 15%                         | $\( 0.67 \times 0.15 = 0.1005 \)$   |
>
> ### **2. SER Compute Costs**
>
> In addition to the reduced human-labeling cost, the SER method incurs compute costs for self-labeling.
>
> **GPU** **Pricing:** AWS cost for 8 A100 GPUs: $32.77/hour.  **Inference** **Throughput****:** 1,530 examples processed in 3 minutes on a single A100 GPU.  **Inference Cost per Sample:**$\( \text{Cost} = \frac{32.77/8 \, \text{USD/hour}}{1,530 \, \text{examples/3 minutes} \times 20 \, \text{3-minute slots/hour}} = 0.00001338 \, \text{USD/sample}\)$
>
> In our method, 1 inference is conducted per iteration, resulting in a total of 3 additional inferences.
>
> | **Method** | **Human Label Cost (****USD****)** | **Compute Cost (****USD****)** | **Total Cost** **(****USD****)** |
> | ---------- | ---------------------------------- | ------------------------------ | -------------------------------- |
> | SER Method | 0.1005                             | 0.00001338 * 3                 | 0.10054                          |
>
> ### **3.** **LLM** **Labeling Costs**
>
> We evaluated replacing 15% of human labels in the SER method with GPT-4o annotations. For a preference pair, each response is scored 3 times by GPT-4o.
>
> - **Input** **Tokens:** Average input length: 525 tokens.  **Output Tokens:** Average output length: 104 tokens.
> - **GPT-4o Costs:**
>   - Input: $0.0025 per 1,000 tokens.
>   - Output: $0.01 per 1,000 tokens.
> - **Cost Calculation:**$\(\text{Cost per Example} = 6 \times \left( \frac{525 \times 0.0025}{1,000} + \frac{104 \times 0.01}{1,000} \right) = 0.0135 \, \text{USD/sample}\)$
>
> | **Method**            | **LLM** **Label Cost (****USD****)** | **Compute Cost (****USD****)** | **Total Cost** **(****USD****)** |
> | --------------------- | ------------------------------------ | ------------------------------ | -------------------------------- |
> | 15% LLM-Labeled + SER | 0.0135 *0.15                         | 0.00001338 * 3                 | 0.002                            |
>
> ## **Performance and Cost Comparison**
>
> | **Method**              | **Accuracy (%)** | **Cost per Sample (****USD****)** |
> | ----------------------- | ---------------- | --------------------------------- |
> | Full Dataset            | 70.45            | 0.67                              |
> | 15% Human-Labeled + SER | 68.56            | 0.10051                           |
> | 15% LLM-Labeled + SER   | 67.64            | 0.002                             |
>
> **We observe that:**
>
> 1. **Cost-Effectiveness of** **SER****:**  The SER method achieves performance closely aligned with fully human-labeled data while significantly reducing costs.  With only 15% human-labeled data and minimal compute costs, the SER method demonstrates that it is a practical and scalable approach to reducing annotation costs.
> 2. **Self-Improvement Capability:**  The SER method can effectively self-improve using both human-labeled and LLM-labeled data, enabling flexible integration of diverse labeling strategies.
> 3. **Trade-Off Between Annotation and Compute Costs:**  By leveraging accelerators for self-labeling, SER minimizes annotation costs without incurring prohibitive compute costs.  At \( K = 4 \) loops, SER achieves a strong balance between performance and cost, suggesting it is a viable alternative to full human labeling for various tasks.
> 4. **Scalability:**
>    1. The SER method provides a cost-efficient framework for scaling reward models without significantly compromising performance, making it suitable for real-world applications where annotation budgets are constrained.

---

> > ### Comment · Reviewer_8P3z · 2024-11-24
> > **Thank you for the response**
> >
> > Thank you for the well articulated response! From the computations, it seems the conclusion is the proposed method is more effective, quality being close for a fraction of the cost. I have a follow-up question.
> >
> > *Can the (15% LLM-labeled + SER) outperform the "Full dataset" accuracy if using the same cost basis? Why or why not*
> >
> > *Note:* I am not expecting the authors to run further experiments, I am looking for evidence and intuition as reader and practitioner.

---

> > > ### Author Response · Authors · 2024-11-26
> > > **Thank you for your valuable feedback**
> > >
> > > Thank you for your valuable feedback and the opportunity to address your concerns. We hope our responses have provided clarity and further strengthened the work. We would sincerely appreciate your consideration of a higher score.

---

> ### Author Response · Authors · 2024-11-25
> **Response to Comment2**
>
> **Comment2**
>
> Thank you for the well articulated response! From the computations, it seems the conclusion is the proposed method is more effective, quality being close for a fraction of the cost. I have a follow-up question.
>
> *Can the (15% LLM-labeled + SER) outperform the "Full dataset" accuracy* *if using the same cost basis**? Why or why not**？*
>
> **Response to Comment2:** Thank you for your thoughtful question. Let's explore whether the (15% LLM-labeled + SER) method can outperform the "Full Dataset" accuracy when using the same cost basis.
>
> Refer to comment 1, '15% LLM-labeled+SER' cannot outperform 'Full dataset'. However, it is hard to make the cost similar to '15% LLM-labeled+SER' with more SER loops as there is less data filtered for later loop training. Then we increase the percentage of the labeled data to increase the cost. Comparing with '15% LLM-labeled+SER' and '15% human-labeled+SER', the latter has 68.56 acc with $0.1005 cost, better performance with more cost. Hence we use '60% human-labeled+SER' with similar cost to 'Full dataset' with the results in the below table:
>
> ### Full Dataset Method:
>
> - **Accuracy:** 70.45
> - **Cost per Sample:** \$0.67
>
> ### 60% Humam-labeled +SER Method
>
> - **Accuracy**: **71.83**
> - Cost per Sample： \$0.402
>
> ### 15% Human-labeled +SER Method
>
> - **Accuracy**: 68.56
> - **Cost per sample**： \$0.10051
>
> Under lower cost, the （60% human labeled + SER) outperform the "Full Dataset" accuracy due to: （1）**Curriculum Learning with SER**: employs a curriculum learning approach, guiding the model from easier tasks (distinguishing good vs. bad responses) to harder tasks (discerning subtle differences). This progression allows the model to build a solid foundation before tackling complex distinctions, leading to more effective learning overall. **(2) Reinforcement of Correct Patterns**: By iteratively training on high-confidence self-labeled data, the model reinforces accurate patterns and refines its decision boundaries. This results in improved performance compared to training solely on human-labeled data, as the model effectively learns from the most informative examples.**(3) Amplification of Informative Data**: Combining human-labeled data with SER allows the model to leverage additional high-quality, self-labeled examples, increasing the diversity and richness of the training set without requiring full human annotation. **(4) Mitigation of Overfitting**: Training exclusively on 100% human-labeled data can lead to overfitting or bias toward the annotated examples. Incorporating SER introduces variability that helps the model generalize better to unseen data by reducing reliance on potentially biased human annotations.

---

> ### Author Response · Authors · 2024-12-03
>
> We greatly appreciate your insightful feedback and suggestions.  As the discussion period concludes, we trust that our detailed responses have effectively addressed your concerns and showcased the enhancements made to strengthen the paper.  We aim for these improvements to meet your expectations and align with the criteria for a higher score.

---

### Author Response · Authors · 2024-11-28
**Global response**

We sincerely thank all reviewers [R1 (8P3z), R2 (vMDv), R3 (U5TR), R4 (eZmA)] for their thoughtful and constructive feedback. We are delighted that the reviewers recognized the importance of the problem addressed [R1, R2], appreciated the well-supported motivation [R1, R2, R3], and found our experiments comprehensive with promising results [R1, R2, R3, R4].

Based on the reviewers' feedback, we have carefully revised the manuscript and uploaded a new version for review. All changes are highlighted in blue. Below is a summary of the key updates:

- **Section 3.1**: We provided a clearer explanation of Equation 3 and elaborated on the rationale behind its design [R2, R3, R4].
- **Appendix A**: A more rigorous theoretical analysis and proof have been added to strengthen the technical rigor of the paper [R2].
- **Appendix B**: The pseudo-code has been updated to better clarify the training process, incorporating feedback to address potential misunderstandings [R3, R4].
- **Appendix C.3**: The hyper-parameter tuning process has been updated to enhance the clarity [R2,R4].
- **Appendix E**: The cost of SER and the human-labeled methods has been updated, addressing the reviewers' requests for more detailed comparisons [R1].
- **Section 4.1.1**: Sentences were revised to improve clarity and avoid confusion, aligning with reviewers’ suggestions [R2, R3].
- Typos and minor errors throughout the manuscript have been corrected.

We deeply appreciate the reviewers’ insights, which have significantly enhanced the quality and presentation of our work. Thank you again for your thoughtful and valuable feedback.

---

### Meta-Review · Area_Chair_RwXA · 2024-12-19

**Metareview:**

The paper presents a new method to have a reward model to generate more data to "cover for its weaknesses". This is reminiscent of much work done in effective RL exploration but from the perspective of improving a learned human proxy reward instead of a policy. The idea is sound and well situated with the existing literature.

Many reviewer's raised questions initially about the validity of the claims made in light of the fact that only preference accuracy was reported in the evals which is rather non standard. After rather extensive discussions with the reviewers, the authors appear to have convinced a majority of them (and me) that the paper is suitable for publication by providing additional results on more indicative experimental settings. I'd highly encourage the authors to make the changes the reviewers especially those regarding clarity that vMDv and U5TR suggest and to include all of these results in the paper itself.

**Additional Comments On Reviewer Discussion:**

eZmA is the only one who recommended rejection after the rebuttal phase though they did not engage with the authors. It appear to me that many of the reviewer's concerns were addressed. Other reviewers also increased their scores until 3/4 recommend acceptance.

---

### Decision · Program_Chairs · 2025-01-22

Accept (Poster)